# Transcriptional and epigenomic landscapes of CNS and non-CNS vascular endothelial cells

Mark F Sabbagh[1,2], Jacob S Heng[1,2], Chongyuan Luo[3,4], Rosa G Castanon[3], Joseph R Nery[3], Amir Rattner[1], Loyal A Goff[2,5], Joseph R Ecker[3,4], Jeremy Nathans[1,2,6,7]*

[1]Department of Molecular Biology and Genetics, Johns Hopkins University School of Medicine, Baltimore, United States; [2]Department of Neuroscience, Johns Hopkins University School of Medicine, Baltimore, United States; [3]Genomic Analysis Laboratory, The Salk Institute for Biological Studies, La Jolla, United States; [4]Howard Hughes Medical Institute, The Salk Institute for Biological Studies, La Jolla, United States; [5]Institute for Genetic Medicine, Johns Hopkins University School of Medicine, Baltimore, United States; [6]Department of Ophthalmology, Johns Hopkins University School of Medicine, Baltimore, United States; [7]Howard Hughes Medical Institute, Johns Hopkins University School of Medicine, Baltimore, United States

**Abstract** Vascular endothelial cell (EC) function depends on appropriate organ-specific molecular and cellular specializations. To explore genomic mechanisms that control this specialization, we have analyzed and compared the transcriptome, accessible chromatin, and DNA methylome landscapes from mouse brain, liver, lung, and kidney ECs. Analysis of transcription factor (TF) gene expression and TF motifs at candidate *cis*-regulatory elements reveals both shared and organ-specific EC regulatory networks. In the embryo, only those ECs that are adjacent to or within the central nervous system (CNS) exhibit canonical Wnt signaling, which correlates precisely with blood-brain barrier (BBB) differentiation and *Zic3* expression. In the early postnatal brain, single-cell RNA-seq of purified ECs reveals (1) close relationships between veins and mitotic cells and between arteries and tip cells, (2) a division of capillary ECs into vein-like and artery-like classes, and (3) new endothelial subtype markers, including new validated tip cell markers.
DOI: https://doi.org/10.7554/eLife.36187.001

*For correspondence:
jnathans@jhmi.edu

## Introduction

In vertebrates, the vascular and immune systems constitute the only classes of cells that come into close proximity to nearly all other cell types in the body. These systems also share the property that a small number of basic cell types are further divided into functionally and molecularly distinct subtypes, some of which are specialized for the tissues in which they reside. For example, in the immune system, phagocytic cells are represented by monocytes in the blood, Kupffer cells in the liver, microglia in the central nervous system (CNS), Langerhans cells in the skin, and osteoclasts in bone. In the vascular system, organ-specific molecular and cellular specialization is typically at the level of microvascular (capillary) endothelial cells (ECs) (*Aird, 2007a*; *Aird, 2007b*; *Potente and Mäkinen, 2017*). Additional specializations, which apply across all tissues, distinguish arteries, veins, and capillaries.

Liver and brain capillaries exemplify two extremes of EC specialization. The highly permeable sinusoidal vascular endothelium of the liver acts as a sieve in which ECs form a discontinuous lining to allow efficient diffusional exchange between serum and hepatocytes. Hepatic capillaries also

exhibit one of the body's highest capacities for endocytosis through the expression of various classes of scavenger receptors (*Aird, 2007b*; *Knolle and Wohlleber, 2016*; *Poisson et al., 2017*). In contrast, the vast majority of capillaries in the CNS have extremely low permeability. CNS ECs are characterized by reduced transcytosis, an absence of fenestrations, tight junctions between neighboring cells, and a variety of small molecule transporters and efflux pumps (*Zhao et al., 2015*). Together, these properties constitute the blood-brain barrier (BBB).

The widespread functional, morphological, and molecular heterogeneity of ECs must ultimately reflect different gene expression programs. The BBB program represents a particularly interesting case, as it involves the coordinated regulation of a large number of genes (*Daneman et al., 2010*). Early transplantation studies showed that the BBB properties of CNS ECs are induced by developing neural tissue (*Stewart and Wiley, 1981*), implicating an instructive signal from surrounding parenchyma. More recent work has shown that WNT and NORRIN ligands produced by neurons and glia activate receptors on ECs, and the resulting canonical Wnt signal initiates and maintains the BBB state (*Liebner et al., 2008*; *Stenman et al., 2008*; *Daneman et al., 2009*; *Wang et al., 2012*; *Zhou et al., 2014*; *Cho et al., 2017*). Interestingly, the same canonical Wnt signal acts earlier in vascular development to direct CNS angiogenesis.

Most molecular studies of EC heterogeneity have focused on defining transcriptome differences between EC subtypes (*Chi et al., 2003*; *Daneman et al., 2010*; *Nolan et al., 2013*; *Hupe et al., 2017*). Several recent studies have also investigated the enhancer landscape of ECs (*Ernst et al., 2011*; *Hogan et al., 2017*; *Quillien et al., 2017*; *Zhou et al., 2017*). However, there have been no genome-wide analyses that integrate and compare the transcriptome, chromatin structure, and DNA methylation landscapes of ECs. Accessible chromatin and genomic regions with low levels of CG methylation are of particular interest as they mark candidate *cis*-regulatory elements (CREs) (*Stadler et al., 2011*; *Thurman et al., 2012*; *Stergachis et al., 2013*; *Ziller et al., 2013*). Importantly, CG hypomethylation can also provide a record of a cell's developmental history, offering an opportunity to identify lineage-specific regulators (*Hon et al., 2013*; *Mo et al., 2015*; *Mo et al., 2016*).

Here, we explore transcriptome, chromatin, and DNA methylation differences among four populations of murine ECs to determine the factors that are associated with EC heterogeneity. We uncover unique sets of candidate CREs that define different EC subtypes, and we identify candidate transcription factor (TF) regulators using motif enrichment analysis of these regions. We provide evidence for precisely spaced and oriented ETS and ZIC TF motifs that likely play a role in defining the enhancer landscape of brain ECs. To explore intra-tissue EC heterogeneity, we apply RNA profiling of brain ECs at the single-cell level, leading to a molecular definition of the relationship among the major CNS EC subtypes. Together, these studies provide an integrated view of the genetic and epigenetic mechanisms underlying the unique tissue-specific programs of EC differentiation.

## Results

### Tissue-specific endothelial cell gene expression

To characterize the genetic basis of inter-tissue heterogeneity among ECs, we set out to acquire a broad view of their genomic and epigenomic differences. Toward this end, we isolated ECs from four different murine tissues - brain, liver, lung, and kidney - at postnatal day 7 (P7) and performed RNA-seq, MethylC-seq (*Lister et al., 2008*), and ATAC-seq (*Buenrostro et al., 2013*) to generate an atlas of transcript abundances, single-base resolution DNA methylation, and accessible chromatin, respectively (*Supplementary file 1*). The data are available for public exploration as the Vascular Endothelial Cell Trans-omics Resource Database (VECTRDB) at https://markfsabbagh.shinyapps.io/vectrdb/ and https://github.com/mfsabbagh/EC_Genomics. To visualize the data on a genome browser, see http://neomorph.salk.edu/Endothelial_cell_methylome.php. At P7, the vasculature is still growing but it has also, presumably, acquired tissue-specific properties to support organ function. To isolate ECs, we harvested organs from *Tie2-GFP* (the *Tie2* gene is also known as *Tek*) mice, which specifically express green fluorescent protein (GFP) in ECs (*Motoike et al., 2000*), enzymatically dissociated the tissues into single-cell suspensions, and then purified ECs using fluorescence-activated cell sorting (FACS) (*Figure 1—figure supplement 1A*). This approach to EC isolation was optimized and validated by *Daneman et al., 2010*). We note that *Tie2-GFP* is not expressed in all

ECs. For example, we could not detect GFP in the capillaries of renal glomeruli by immunostaining (yellow arrows in *Figure 1—figure supplement 1B*). Each high-throughput sequencing experiment was performed on two or more independent biological replicates, and these exhibited high pairwise correlations (*Figure 1A* and far right panel of *Figure 1B*; *Figure 1—figure supplements 2*, *3* and *4A*; *Figure 2A*; *Figure 2—figure supplement 1A*).

To filter out non-EC transcripts that derive from lysed parenchymal cells, we also conducted RNA-seq on GFP-negative FACS-sorted cells, and, for completeness, total dissociated tissue. By comparing the expression profiles of GFP-positive cells to GFP-negative cells, we determined a union set of 4117 protein-coding and non-coding EC-enriched transcripts from each tissue for further analysis (*Supplementary file 2*; see the RNA-seq section under Materials and methods for the three criteria applied in this comparison). Importantly, this list contains many known EC markers, including *Pecam1*, *Kdr*, *Cdh5*, and *Tek* (also known as *Tie2*). Notably absent from the list are the pan-leukocyte marker *Ptprc/Cd45*, the macrophage/microglia marker *Csf1r*, the vascular smooth muscle cell marker *Acta2*, the pericyte marker *Cspg4*, and various abundant parenchyma-specific markers such as *Syt11* (brain), *Alb* (liver), *Sftpc* (lung), and *Umod* (kidney) (*Figure 1—figure supplement 2B*). The GFP-negative fraction of dissociated liver was unexpectedly depleted for hepatocyte-specific transcripts suggesting that hepatocytes were lost during cell dissociation and/or flow cytometry. This fraction was enriched for transcripts associated with Kupffer cells (*Cd68*, *Cd80*, and *Cd86*) and cholangiocytes (*Epcam*, *Pkhd1l1*, and *St14*; *Li et al., 2017*) (*Figure 1—figure supplement 3A*). Based on the retention of known EC-specific transcripts and the removal of known parenchyma-specific transcripts, we conclude that the EC-enrichment criteria selected for bona fide EC-specific transcripts.

Focusing only on the 3853 protein-coding EC-enriched transcripts, scatter plots of between-sample normalized RNA-seq read counts for ECs from one tissue versus the three other EC subtypes show both previously identified and novel tissue-specific EC markers (*Figure 1B*; *Figure 1—figure supplement 4B*). In this analysis, the relatively low correlation between different tissue EC transcriptomes reflects the exclusion of housekeeping genes. Brain ECs specifically express mRNAs coding for transporters such as *Mfsd2a*, *Slc2a1*, and *Slco1c1* (*Figure 1B*, first three panels), whereas liver sinusoidal ECs specifically express mRNAs coding for scavenger receptors such as *Fcgr2b*, *Stab2*, and *Clec4g* (*Figure 1B*, first panel). Similarly, mRNA coding for angiotensin-converting enzyme (*Ace*), a known lung EC marker, was specifically enriched in lung ECs (*Figure 1B*, second panel). The differential expression analysis also identified two novel lung EC markers, *Scn7a* and *Scn3b* (*Figure 1B*; *Figure 1—figure supplements 3B* and *4B*). The former is a sodium channel referred to as Na(x) that is responsive to extracellular levels of sodium and that was previously reported to be expressed in lung epithelium (*Watanabe et al., 2000*; *Watanabe et al., 2002*; *Xu et al., 2015*), while the latter is an auxiliary beta subunit that can modify the conductance of sodium channel alpha subunits (*Cusdin et al., 2010*). *Ctfr*, the cystic fibrosis transmembrane conductance regulator, and surfactant proteins *Sftpa1*, *Sftpb*, *Sftpc*, and *Sftpd* exhibit enrichment in the lung GFP-negative fraction, consistent with a known lung epithelial role for these genes (*Figure 1—figure supplement 3B*). In contrast, both *Scn7a* and *Scn3b* show a clear enrichment in lung ECs on par with *Ace* enrichment, implying that they function in ECs rather than in the epithelium (*Figure 1—figure supplement 3B*). We speculate that expression of *Scn7a* and *Scn3b* may allow lung ECs to directly sense serum sodium concentration, perhaps to regulate *Ace* expression and thereby provide feedback control of the renin-angiotensin-aldosterone system.

Transcripts from 2392 genes exhibited >2 fold differential abundance between one or more pairs of EC subtypes, with 739 exhibiting >2 fold differential expression in a comparison of ECs from one tissue versus ECs from each of the other three tissues (*Figure 1C*; *Supplementary file 2*; see Materials and methods). We will refer to these 739 genes as 'EC-tissue-specific genes' or 'ECTSGs'. For reference, this abbreviation and those that follow are summarized and defined in *Table 1*. Of the 739 ECTSGs, 685 are protein-coding and 54 are non-coding. Gene ontology (GO) enrichment analysis (*Ashburner et al., 2000*; *The Gene Ontology Consortium, 2017*; *The Gene Ontology Consortium, 2017*) of the four sets of ECTSGs was consistent with tissue-specific specializations of each vascular bed. For example, brain ECTSGs were enriched for GO terms such as 'neutral amino acid transport' and 'drug transmembrane transport' whereas liver ECTSGs were enriched for GO terms such as 'scavenger receptor activity' and 'vesicle-mediated transport' (*Supplementary file 3*). Consistent with the known role of Wnt signaling in brain endothelium development, brain ECTSGs were also

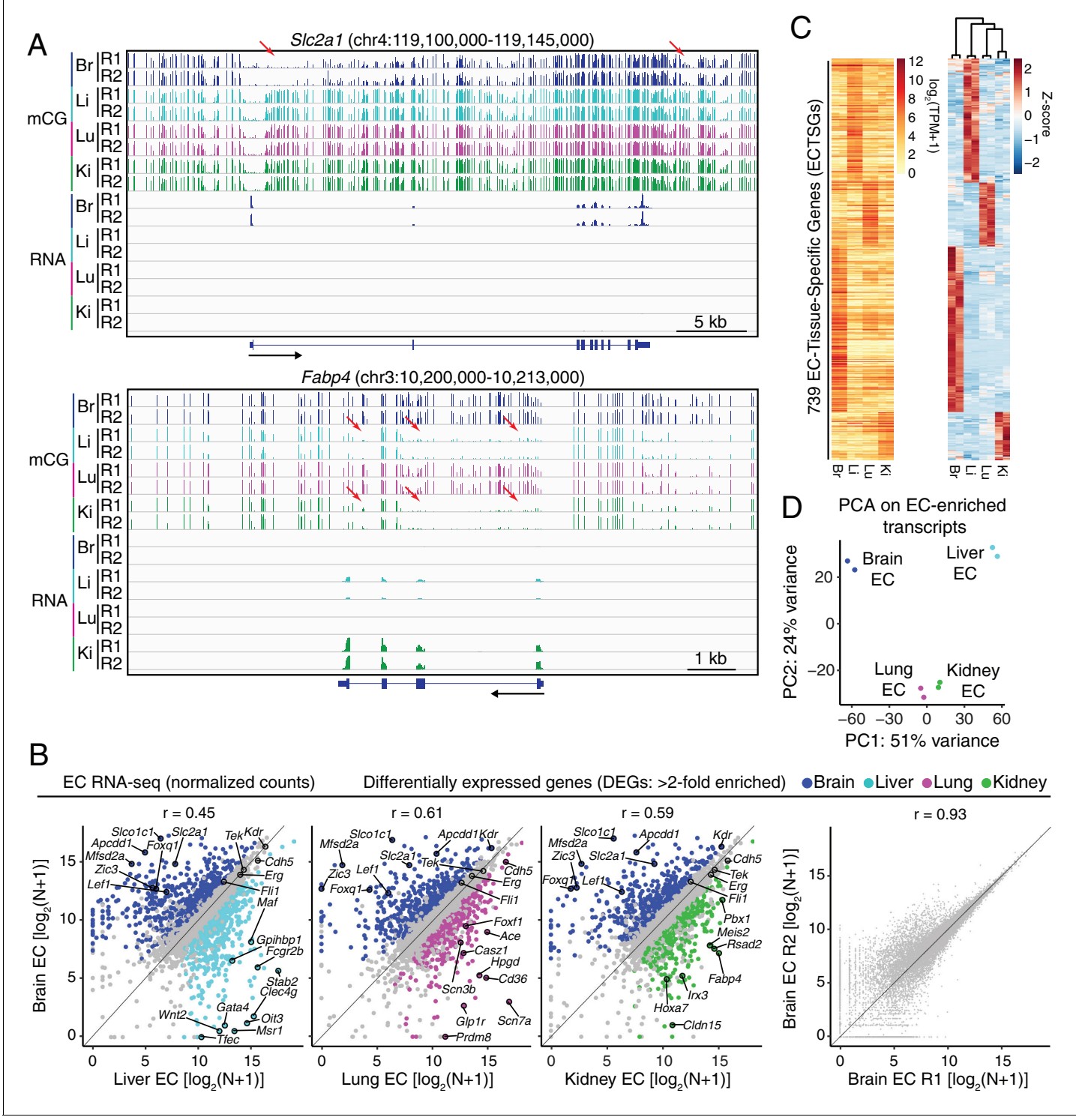

**Figure 1.** RNA-seq reveals inter-tissue EC heterogeneity. (**A**) Genome browser images showing CG methylation (top) and RNA expression (bottom) for two genes: *Slc2a1*, a glucose transporter expressed in brain ECs, and *Fabp4*, a fatty acid binding protein expressed in liver and kidney ECs. For DNA methylation, the mCG/CG ratio is shown, with the height of each bar indicating the fractional methylation (range: 0 to 1). For RNA-seq, histograms of the number of aligned reads are shown. For this and all other genome browser images, the heights of all the tracks of a given sequencing experiment are the same across samples. For both genes, tissue-specific gene expression is associated with tissue-specific hypomethylation near the TSS. Red arrows indicate illustrative examples of differential hypomethylation. Br, brain; Li, liver; Lu, lung; Ki, kidney. R1 and R2, biological replicates. Black arrows beneath this and all other genome browser images indicate the direction of transcription. (**B**) Scatter plots comparing cross-sample normalized RNA-

*Figure 1 continued on next page*

*Figure 1 continued*

seq read counts of EC-expressed protein-coding genes from brain versus liver, lung, and kidney, showing only those transcripts with TPM >10 for each of the two RNA-seq replicates. Colored symbols indicate transcripts with FDR < 0.05 and enrichment >2 fold for the indicated tissue comparison. Right, comparison of cross-sample normalized RNA-seq read counts for protein-coding genes between the two brain EC replicates. Values depicted are the log$_2$ transformation of cross-sample normalized counts + 1. (C) Heatmaps depicting transcript abundances for 739 Endothelial Cell Tissue-Specific Genes (ECTSGs). Left, log$_2$ transformation of TPM +1. Right, z-scores for the TPMs. (D) Principal component analysis of all EC-enriched transcripts from brain, liver, lung, and kidney. The two symbols for each sample represent biological replicates. In this and all other figures, tissue type is indicated by color: Br, brain, blue; Li, liver, cyan; Lu, lung, magenta; Ki, kidney, green.

DOI: https://doi.org/10.7554/eLife.36187.002

The following figure supplements are available for figure 1:

**Figure supplement 1.** *Tie2-GFP* transgenic mouse enables isolation of ECs.

DOI: https://doi.org/10.7554/eLife.36187.003

**Figure supplement 2.** GFP-positive FACS-sorted cells from P7 *Tie2-GFP* mice represent pure populations of ECs.

DOI: https://doi.org/10.7554/eLife.36187.004

**Figure supplement 3.** Expression of cell-type-specific transcripts in liver and lung samples.

DOI: https://doi.org/10.7554/eLife.36187.005

**Figure supplement 4.** RNA-seq comparisons between peripheral ECs.

DOI: https://doi.org/10.7554/eLife.36187.006

enriched for the GO term 'negative regulation of canonical Wnt signaling pathway' (*Supplementary file 3*). Principal component analysis (PCA) of the four EC transcriptomes showed that the transcriptional signatures of brain ECs and liver ECs are substantially distinct from each other and from those of lung ECs and kidney ECs (*Figure 1D*), broadly consistent with earlier studies of embryonic and adult ECs (*Daneman et al., 2010*; *Hupe et al., 2017*).

## Identification of distinct classes of hypomethylated regions in endothelial cells

Low cytosine DNA methylation (mCG) immediately downstream of transcriptional start sites (TSSs) typically correlates with increased gene expression (*Schultz et al., 2015*). In visually comparing EC methylation profiles to transcription profiles, we find that differentially expressed genes are often associated with tissue-specific hypomethylation starting at the TSS and extending into the gene body (*Figure 1A*). As a result, plotting the mean fraction of mCG methylation in a five kilobase (kb) window downstream of the TSS for all protein-coding genes reveals an inverse relationship between methylation and gene expression among differentially expressed EC-enriched genes (*Figure 2A*; *Figure 2—figure supplement 1B*). However, these differences in methylation at differentially expressed genes occur in the context of highly similar global methylation patterns, as seen by correlation coefficients between different EC subtypes (0.95–0.97) that are nearly indistinguishable from the correlation coefficients between biological replicates (0.95–0.98; *Figure 2A*; *Figure 2—figure supplement 1A*).

The distinctive tissue-specific patterns of ECTSGs imply that ECs regulate transcription with a corresponding set of tissue-specific CREs. [Although the use of the term 'CRE' implies that these non-coding genomic regions influence nearby genes (i.e. act in cis), we recognize that some of these genetic elements may act between chromosomes (i.e. in trans).] To identify candidate EC CREs based on patterns of DNA methylation, we first parsed individual EC DNA methylomes into unmethylated regions (UMRs; hypomethylated, CG-rich) and low-methylated regions (LMRs; hypomethylated, CG-poor). These criteria were based on those of *Stadler et al. (2011)* and *Burger et al., 2013*, with some modifications (e.g. methylation threshold $m < 0.3$; see Materials and methods) to account for a relatively low level of global CG methylation (mean of 67%). We identified 16,158 brain EC, 16,506 liver EC, 16,171 lung EC, and 15,979 kidney EC UMRs (median length = 1738 bp; median mCG = 10%), and 72,061 brain EC, 60,212 liver EC, 57,594 lung EC, and 71,343 kidney EC LMRs (median length = 490 bp; median mCG = 15%) (*Supplementary file 4*). 72% of the UMRs (12,581 of 17,406) are within 2 kb of a TSS whereas only 5.7% of the LMRs (6682 of 116,273) are within 2 kb of a TSS (*Figure 2—figure supplement 1C*), consistent with an earlier study showing that UMRs are enriched at promoters and LMRs are enriched at distal regulatory elements (*Stadler et al., 2011*).

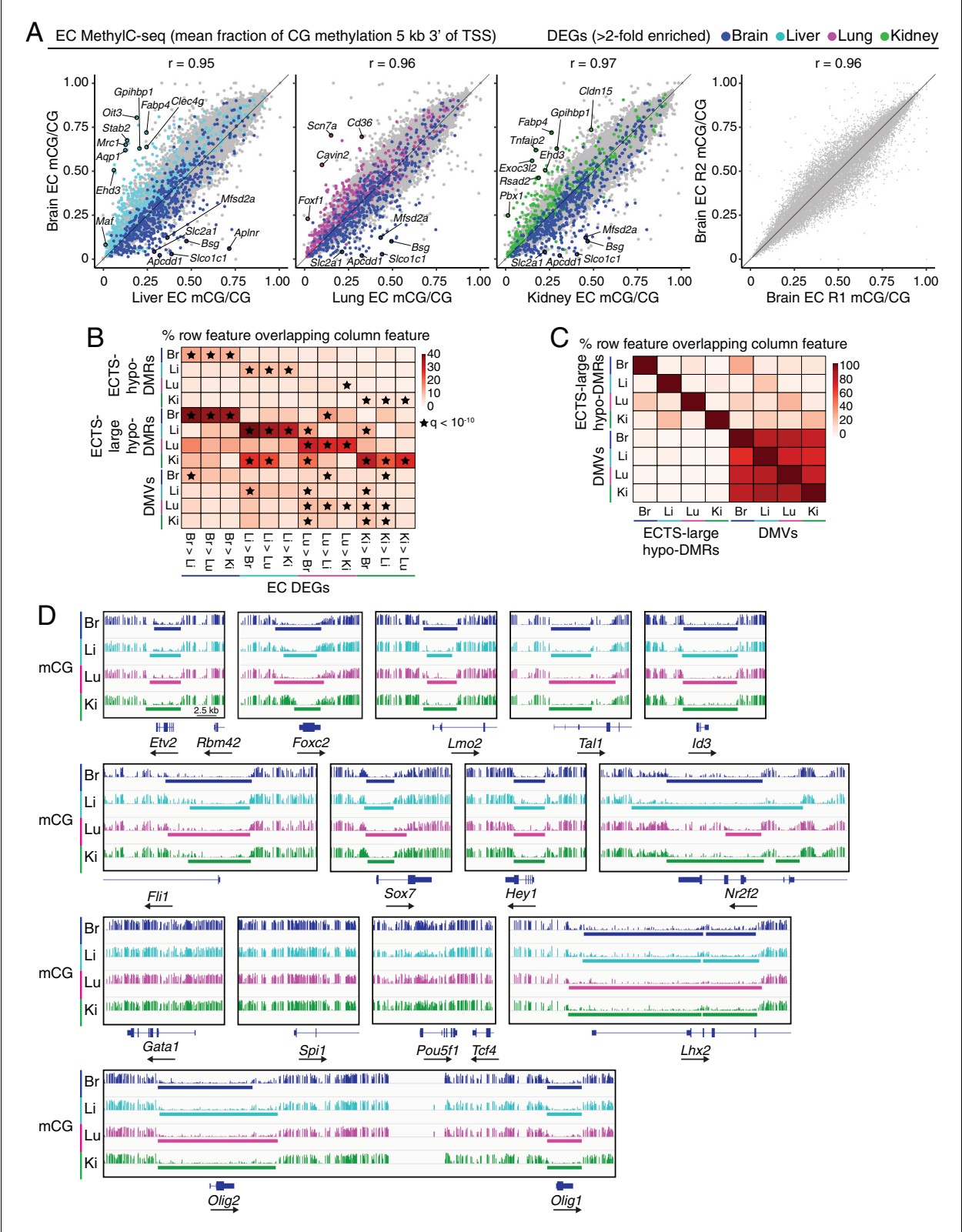

**Figure 2.** MethylC-seq reveals distinct classes of hypomethylated regions in ECs. (**A**) Scatter plots comparing the mean fraction of CG methylation in a 5 kb window immediately 3' of the TSS for protein-coding genes from brain ECs versus liver, lung, and kidney ECs. Colored symbols indicate transcripts with FDR < 0.05 and enrichment >2 fold for the indicated tissue comparison. These plots show a positive correlation between EC tissue-specific gene expression and tissue-specific hypo-methylation. Right, comparison of CG methylation between the two brain EC biological replicates. (**B**) Heatmap

*Figure 2 continued on next page*

*Figure 2 continued*

indicating the percentage of each row that overlaps with differentially expressed genes. A significant proportion of ECTS-large hypo-DMRs overlap differentially expressed genes for the same EC-subtype relative to the other EC subtypes. Black stars indicate statistical significance at q < 1×10$^{-10}$. (**C**) Heatmap indicating the percentage of each row that overlaps with either ECTS-large hypo-DMRs or DMVs. DMVs exhibit more overlap between EC subtypes than large hypo-DMRs. (**D**) Genome browser images showing methylation as in *Figure 1A* at various TF genes. Colored bars indicate DMVs. Each genome browser image is at the same scale.

DOI: https://doi.org/10.7554/eLife.36187.007

The following figure supplements are available for figure 2:

**Figure supplement 1.** Distinct classes of hypomethylated features in ECs.

DOI: https://doi.org/10.7554/eLife.36187.008

**Figure supplement 2.** DMVs at differentially expressed TF genes exhibit differential methylation.

DOI: https://doi.org/10.7554/eLife.36187.009

86% of UMRs (15,036 of 17,406) are shared by all four EC subtypes, whereas only 25% of LMRs (29,053 of 116,273) are shared by all four EC subtypes (*Figure 2—figure supplement 1D*).

We next searched for regions of the genome that were differentially hypomethylated in one or more pairwise comparisons among the four EC subtypes. Unlike the enumeration of UMRs/LMRs, which is an intrinsic property of each DNA methylome, this analysis relies on a comparison between DNA methylomes. Using a conservative statistical approach (*Lister et al., 2013*); see Materials and methods), we discovered 105,050 discrete differentially hypomethylated regions ('hypo-DMRs'; i.e., a % mC difference above threshold in one or more pairwise comparisons between EC tissue types; median length = 228 bp) with the largest number present in liver ECs (42,682), followed by lung ECs (32,004), brain ECs (30,471), and kidney ECs (17,027). A smaller number of hypo-DMRs were uniquely hypomethylated in one EC tissue type (25,612 in liver ECs, 13,862 in lung ECs, 16,540 in brain ECs, and 7132 in kidney ECs). We will refer to these regions as 'EC-tissue-specific hypo-DMRs' or 'ECTS-hypo-DMRs' (*Supplementary file 4*). Almost all hypo-DMRs (94.5%) are found >2 kb from a TSS, suggesting they reflect distal regulatory elements (*Figure 2—figure supplement 1C*). This pattern of EC tissue-type diversity among hypo-DMRs and LMRs implies differential utilization of the enhancer landscape.

**Table 1.** Abbreviations used throughout the text

| Abbreviation | Definition |
| --- | --- |
| ATAC | Assay for Transposase Accessible Chromatin |
| BBB | Blood-brain barrier |
| CRE | *Cis*-regulatory element |
| DEG | Differentially expressed gene |
| DMR | Differentially methylated region |
| DMV | DNA methylation valley (UMR > 3 kb) |
| EC | Endothelial cell |
| ECTS-hypo-DMR | Endothelial cell tissue-specific hypo-DMR |
| ECTSAP | Endothelial cell tissue-specific ATAC-seq peak |
| ECTSG | Endothelial cell tissue-specific gene |
| GO | Gene ontology |
| LMR | Low-methylated region (hypomethylated, CG-poor) |
| PCA | Principal component analysis |
| scRNA-seq | Single-cell RNA-seq |
| TF | Transcription factor |
| TSS | Transcription start site |
| UMR | Unmethylated region (hypomethylated, CG-rich) |

DOI: https://doi.org/10.7554/eLife.36187.010

A previous study in neurons has shown that many closely spaced hypo-DMRs are part of larger hypomethylated regions that correlate with the current transcriptional state of a cell and with higher levels of histone modifications associated with active enhancers (*Mo et al., 2015*). Therefore, we expanded our analysis to assess regions of low DNA methylation that span multiple kilobases. Using an approach similar to the one described by *Mo et al. (2015*), we merged closely spaced (separation <1 kb) hypo-DMRs into larger hypomethylated regions; we refer to such regions as 'large hypo-DMRs' if they are >2 kb in size (*Supplementary file 4*). ECTS-large hypo-DMRs are enriched at differentially expressed EC-enriched genes (q-value <$10^{-10}$; *Figure 2B*; see *Supplementary file 5*).

DNA methylation valleys (DMVs; also referred to as DNA methylation canyons) are a distinct category of large hypomethylated regions that exhibit especially high enrichment at genes coding for important regulators of embryonic development including many TFs (*Xie et al., 2013*; *Jeong et al., 2014*; *Mo et al., 2015*; *Mo et al., 2016*). Defining DMVs as UMRs > 3 kb, we identified 2313 brain EC, 2695 liver EC, 2403 lung EC, and 2208 kidney EC DMVs (*Supplementary file 4*). We found very little overlap between large hypo-DMRs and DMVs, implying they represent distinct hypomethylated features (*Figure 2C*). These two categories of hypomethylation are further distinguished by the extent to which they are shared between EC types: 1602 DMVs are shared by all four EC subtypes and additional DMVs are shared between subsets of EC subtypes, whereas the majority of large hypo-DMRs are specific to a single EC subtype (*Figure 2C*). We used the Genomic Regions Enrichment of Annotations Tool (GREAT) to identify gene ontologies associated with EC DMVs (*McLean et al., 2010*). Consistent with previous studies in other tissues (*Xie et al., 2013*; *Jeong et al., 2014*; *Mo et al., 2015*; *Mo et al., 2016*), EC DMVs are highly enriched for genes encoding TFs (*Figure 2—figure supplement 1E*).

Previously identified DMVs in embryonic stem cells (ESCs) remain hypomethylated after cellular differentiation, with only a small subset demonstrating DNA methylation variations across cell types (*Xie et al., 2013*). These large regions of hypomethylation are maintained by Polycomb complexes that deposit repressive H3K27me3 histone modifications and are thought to support developmental plasticity, as DNA methylation may reflect a more stable repressive mechanism (*Li et al., 2018*). Interestingly, a subset of H3K27me3-positive DMVs found at lineage-specific expressed genes exhibit increased DNA methylation outside of the promoter region with a corresponding loss of H3K27me3, providing a snapshot of the developmental history of gene expression (*Li et al., 2018*). In all four subtypes of ECs, DMVs overlap *Etv2*, *Foxc2*, *Tal1*, *Lmo2*, *Id3*, and *Fli1*, genes coding for TFs that are important for establishing the EC lineage (*Figure 2D*). DMVs are also present at *Sox7*, *Hey1*, and *Nr2f2*, TFs important for venous or arterial differentiation. An exception to this pattern is seen with the early embryonic TF gene *Pou5f1/Oct4*, which lacks a DMV, consistent with a previous report that this locus exhibits increased DNA methylation upon cell differentiation (*Xie et al., 2013*). No EC DMVs overlap *Gata1* and *Spi1*, which are important for establishing the erythroid and myeloid/lymphoid lineages, respectively. Given that ECs and hematopoietic cells share the same developmental lineage, these observations suggest that DMVs occur at TFs that determine very early progenitor cell lineage commitments. Curiously, EC DMVs overlap some genes coding for lineage-determining TFs in completely unrelated cell types, including oligodendrocytes (*Olig1* and *Olig2*) and Muller glia (*Lhx2*) (*Figure 2D*).

At genes coding for TFs that are differentially expressed among EC subtypes, DMVs show EC subtype-specific differences in boundary locations and degree of methylation, with a broad trend of increased methylation 5′ of the TSS in the EC subtype(s) that express the TF gene as seen for *Foxf1*, *Foxf2*, and *Zic3* in brain; *Tbx3* in brain, lung, and kidney; *Tbx2*, *Gata6*, *Meis1*, and *Meis2* in liver, lung, and kidney; *Gata4* in liver; and *Irx3* in kidney (*Figure 2—figure supplement 2A–C*). This observation is consistent with a proposed model in which cell-type-specific expression of important TF regulators of lineage identity results in nearby hypermethylation (*Li et al., 2018*). We note that FOXF1, a lung EC-enriched TF implicated in pulmonary vascular development (*Kalinichenko et al., 2001*; *Cai et al., 2016*; *Dharmadhikari et al., 2016*), represents an exception to this methylation pattern, due to demethylation at the adjacent lung EC-expressed lncRNA gene, *Fendrr* (*Figure 2—figure supplement 2A*).

## HOX gene expression and DNA methylation: correlation with anterior-posterior position

Some of the largest regions of hypomethylation are found in the HOX gene clusters. HOX genes do not exhibit the classic relationship between methylation and gene expression (*Laurent et al., 2010*; *Bock et al., 2012*; *Xie et al., 2013*). Among the four EC subtypes, HOX cluster DNA methylation is lowest in brain ECs, intermediate in liver and lung ECs, and highest in kidney ECs (*Figure 3A and C*; *Figure 3—figure supplement 1*). This pattern of methylation correlates with the anterior/posterior position of the brain, liver, lung, and kidney – as well as with their embryonic primordia - along the body axis: the brain is most anterior, the lungs and liver are intermediate (and are derived from the primitive foregut), and the kidneys are the most posterior (and are derived from the ureteric bud and surrounding mesenchyme). For all four HOX clusters, the ATAC-seq patterns are similar for liver, lung, and kidney ECs, but the peak intensities are greatly reduced in brain ECs (*Figure 3A*; *Figure 3—figure supplement 1*).

In the HOX clusters, there is a roughly co-linear relationship between each gene's position along the chromosome and the anterior/posterior territory along the body axis where that HOX gene is expressed: genes that are named with low numbers (e.g. *Hoxa1*) are expressed more anteriorly and genes that are named with high numbers (e.g. *Hoxa13*) are expressed more posteriorly (*Lewis, 1978*; reviewed in *Papageorgiou, 2012*). Based on the EC subtype-specific RNA-seq data, we found that patterns of HOX gene expression correlate with the anterior/posterior position of each organ and its corresponding EC subtype (*Figure 3B and D*). In the HOX-A cluster, brain ECs express none of the *Hoxa* genes; liver ECs express *Hoxa2–Hoxa5*; lung ECs express *Hoxa2–Hoxa5* and *Hoxa7*; and kidney ECs express *Hoxa5* and *Hoxa7-Hoxa10*, with lower levels of expression in *Hoxa2-Hoxa4*. (*Figure 3B and D*). Interestingly, there does not appear to be a strong relationship between gene expression and gene body methylation (compare top and bottom heatmaps in *Figure 3D*). Rather, like other developmentally important TFs, the trend is that expression of a HOX gene correlates with increased methylation in the neighborhood of the gene body. For example, *Hoxa5* which is expressed in liver, lung, and kidney ECs but not brain ECs exhibits a hypomethylated gene body in all ECs but a hypermethylated promoter region only in peripheral ECs (*Figure 3A*). Similarly, the regions in and adjacent to *Hoxa10* are hypermethylated only in kidney ECs, which are the only ECs in this set that express *Hoxa10* (*Figure 3A*). For each of the four organs, the patterns of EC and parenchymal cell expression of *Hoxa* genes are closely matched (*Figure 3B*). These data confirm and extend two earlier studies: (1) a microarray transcriptome meta-analysis showing a correlation between patterns of HOX gene expression and the tissue-of-origin for a collection of human EC lines (*Toshner et al., 2014*) and (2) a transgenic mouse study showing that *Hoxa3* and *Hoxc11* reporter expression in murine vasculature roughly corresponds to the positional identity specified by the HOX code (*Pruett et al., 2008*). Taken together, the patterns of HOX gene expression and DNA methylation suggest that ECs in different organs 'know' their anterior-posterior position within the body.

## Comparisons between accessible chromatin and hypomethylation landscapes

As a second and complimentary approach to defining candidate EC CREs, we analyzed regions of accessible chromatin. In our initial analysis, we used the full range of ATAC-seq fragment lengths and identified a total of 51,740 discrete regions of accessible chromatin (i.e. ATAC-seq peaks; median length 656 bp) among the four EC subtypes (*Supplementary file 6*). Across replicates, the percent of mapped reads found in called peaks ranged between 16 and 34% (*Figure 4—figure supplement 1A*). Calling peaks on subsamples of total mapped reads for each ATAC-seq replicate indicates that, with this sequencing depth, the data is approaching an asymptote (*Figure 4—figure supplement 1B*). In deriving a consensus EC-tissue-specific peakset, we adopted a conservative approach by requiring each peak to be called in 2/2 or 2/3 replicates (*Figure 4—figure supplement 1C*).

For a quantitative assessment of open chromatin, we restricted our analyses to ATAC-seq fragments of <100 bp, as previous work suggested that these correspond to nucleosome-free regions of the genome (*Buenrostro et al., 2013*). Using <100 bp fragments, we identified a total of 39,987 ATAC-seq peaks (median length 508 bp) among the four EC subtypes (*Supplementary file 6*).

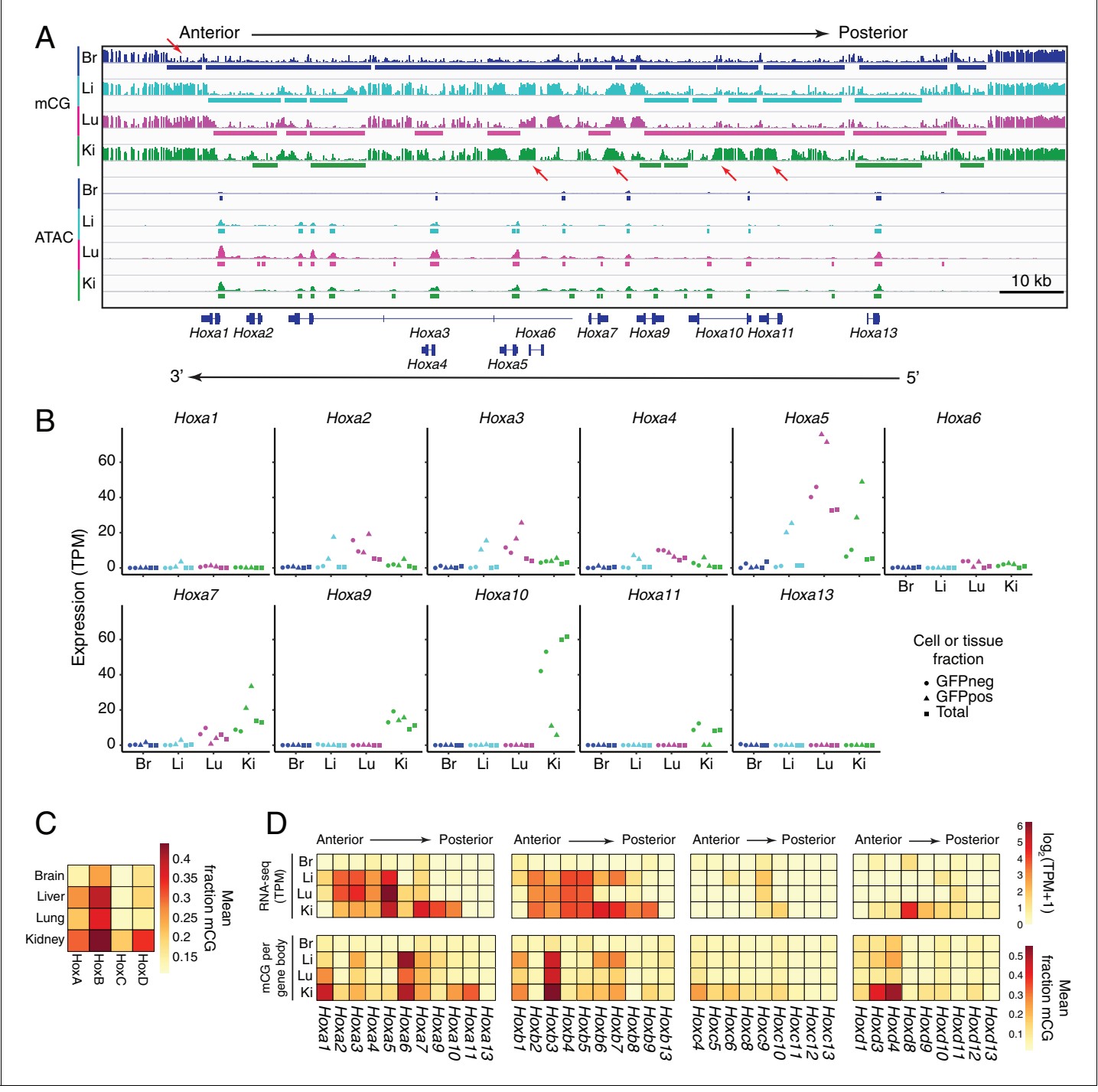

**Figure 3.** In ECs, patterns of methylation and gene expression at HOX gene clusters correlate with anterior/posterior position. (A) Genome browser image showing CG methylation (top) and accessible chromatin (bottom) at the HOX-A gene cluster. HOX genes in this cluster are expressed in an anterior-posterior gradient corresponding to their position in the cluster, with genes near the 3' end of the cluster expressed more anteriorly and genes near the 5' end expressed more posteriorly. The degree of EC methylation is: brain <liver ~ lung<kidney. The degree of EC accessible chromatin is: brain <liver < lung~kidney. Colored bars indicate DMVs or ATAC-seq peaks. Red arrows pointing down indicate illustrative examples of differential hypomethylation. Red arrows pointing up indicate illustrative examples of differential hypermethylation. (B) Expression levels (TPMs) based on RNA-seq for each gene in the HOX-A cluster. (C) Heatmap depicting mean fraction of methylated CG across each HOX cluster for each EC subtype. (D) Heatmaps depicting $\log_2$(TPM +1) or mean fraction of methylated CG in the gene body for genes within each of the four HOX clusters.
DOI: https://doi.org/10.7554/eLife.36187.011

The following figure supplement is available for figure 3:

*Figure 3 continued on next page*

Figure 3 continued

**Figure supplement 1.** Methylation patterns and accessible chromatin at HOX-B, HOX-C, and HOX-D clusters in ECs.
DOI: https://doi.org/10.7554/eLife.36187.012

[Hereafter, the phrase 'ATAC-seq peaks' refers to <100 bp fragments unless otherwise stated.] Of the 31,358 brain EC, 18,926 liver EC, 24,970 lung EC, and 21,694 kidney EC ATAC-seq peaks, 35% are shared across all four EC subtypes, but only 1–16% are EC subtype-specific (6381 brain, 632 liver, 755 lung, and 397 kidney EC-tissue-specific ATAC-seq peaks, referred to hereafter as 'ECT-SAPs') (*Figure 4—figure supplement 2B*).

We observed multiple ECTSAPs near ECTSGs, as illustrated in *Figure 4A* using the full range of ATAC-seq fragment lengths. As indicated by the translucent orange bars in *Figure 4A*, ECTS-hypo-DMRs generally co-localize with ECTSAPs. For example, near brain EC-specific genes *Slc2a1* and *Mfsd2a*, multiple ECTSAPs and ECTS-hypo-DMRs co-localize (*Figure 4A*, upper two panels; *Figure 4—figure supplement 2A*). Similarly, the genomic region encompassing *Clec4g*, a liver EC-specific gene, contains co-localized ECTSAPs and ECTS-hypo-DMRs upstream and downstream of the gene body (*Figure 4A*, right bottom panel; *Figure 4—figure supplement 2A*). A region ~200 kb upstream of *Foxf1* contains two ECTSAPs that colocalize with ECTS-hypo-DMRs (*Figure 4A*, left bottom panel; *Figure 4—figure supplement 2A*), suggestive of a long-range enhancer. This region also includes a lncRNA gene, *Gm26878*, that is expressed specifically in lung ECs (*Figure 4—figure supplement 2A*). In *Figure 4A*, regions of chromatin accessibility and hypomethylation that are shared across more than one EC subtype are indicated by translucent gray bars.

In scatter plots of normalized ATAC-seq read densities, ECTSAPs for each tissue type are enriched near differentially expressed EC-enriched genes for that tissue (*Figure 4B*), and broad differences are apparent between EC subtypes (r = 0.75–0.92), whereas biological replicates of the same subtype are highly similar (r = 0.93–0.97, *Figure 4B*; *Figure 4—figure supplement 3*). Additionally, a larger proportion of ECTSAPs and ECTS-hypo-DMRs are within 100 kb of ECTSGs that are expressed in the corresponding EC subtype relative to ECTSGs expressed in the other EC subtypes, a difference that is highly significant statistically (*Figure 4—figure supplement 2C*). This enrichment implies a role for these candidate CREs in EC subtype-specific gene expression.

Of the 39,987 total ATAC-seq peaks among all four EC subtypes, 64% are >2 kb from a TSS (i.e. are promoter-distal). Similar to the distributions described above for UMRs and LMRs, a majority of promoter-proximal (<2 kb from a TSS) ATAC-seq peaks are shared by all EC subtypes (10,336 of 14,474; 71%), whereas only a small minority of promoter-distal ATAC-seq peaks are shared by all EC subtypes (3476 of 25,517; 14%) (*Figure 4C*, upper two panels; *Figure 4—figure supplement 2D*). This trend is even more striking for methylation at UMRs and LMRs (*Figure 4C*, lower two panels). Chromatin accessibility at UMRs and LMRs and DNA methylation at ATAC-seq peaks similarly shows greater divergence among EC subtypes at distal candidate CREs (*Figure 4—figure supplement 2E*). The ATAC-seq comparisons also reveal a greater degree of similarity in chromatin accessibility among ECs derived from the three peripheral tissue (liver, lung, and kidney) relative to brain ECs (*Figure 4C*). Taken together, these comparisons suggest that the heterogeneity in gene expression that gives rise to tissue-specific EC differentiation may largely reflect differences in distal rather than proximal CREs. As illustrated by the hierarchical clustering analysis shown in *Figure 4C*, these differences are most pronounced in the case of brain ECs, a finding that potentially reflects the specialized and extensive program of gene expression associated with the BBB.

In all four EC subtypes, >80% of ATAC-seq peaks overlap with UMRs or LMRs (*Figure 4—figure supplement 4A*), in agreement with previous findings in neurons and photoreceptors (*Mo et al., 2015*; *Mo et al., 2016*). ATAC-seq peaks from all four EC subtypes exhibit similarly reduced levels of DNA methylation compared to random size-matched genomic regions. Moreover, ECTSAPs exhibit levels of DNA methylation that are preferentially reduced in their respective EC DNA methylomes (*Figure 4—figure supplement 4B*). Reciprocally, there were preferential increases in the EC subtype-specific ATAC-seq signal in the respective ECTS-hypo-DMRs (*Figure 4—figure supplement 4C*).

Interestingly, there are many more LMRs than ATAC-seq peaks, and therefore only a minority of LMRs overlap with ATAC-seq peaks:<10% of liver EC and kidney EC LMRs and <20% of brain and lung EC LMRs overlap with ATAC-seq peaks (*Figure 4—figure supplement 4A*). In contrast, >75%

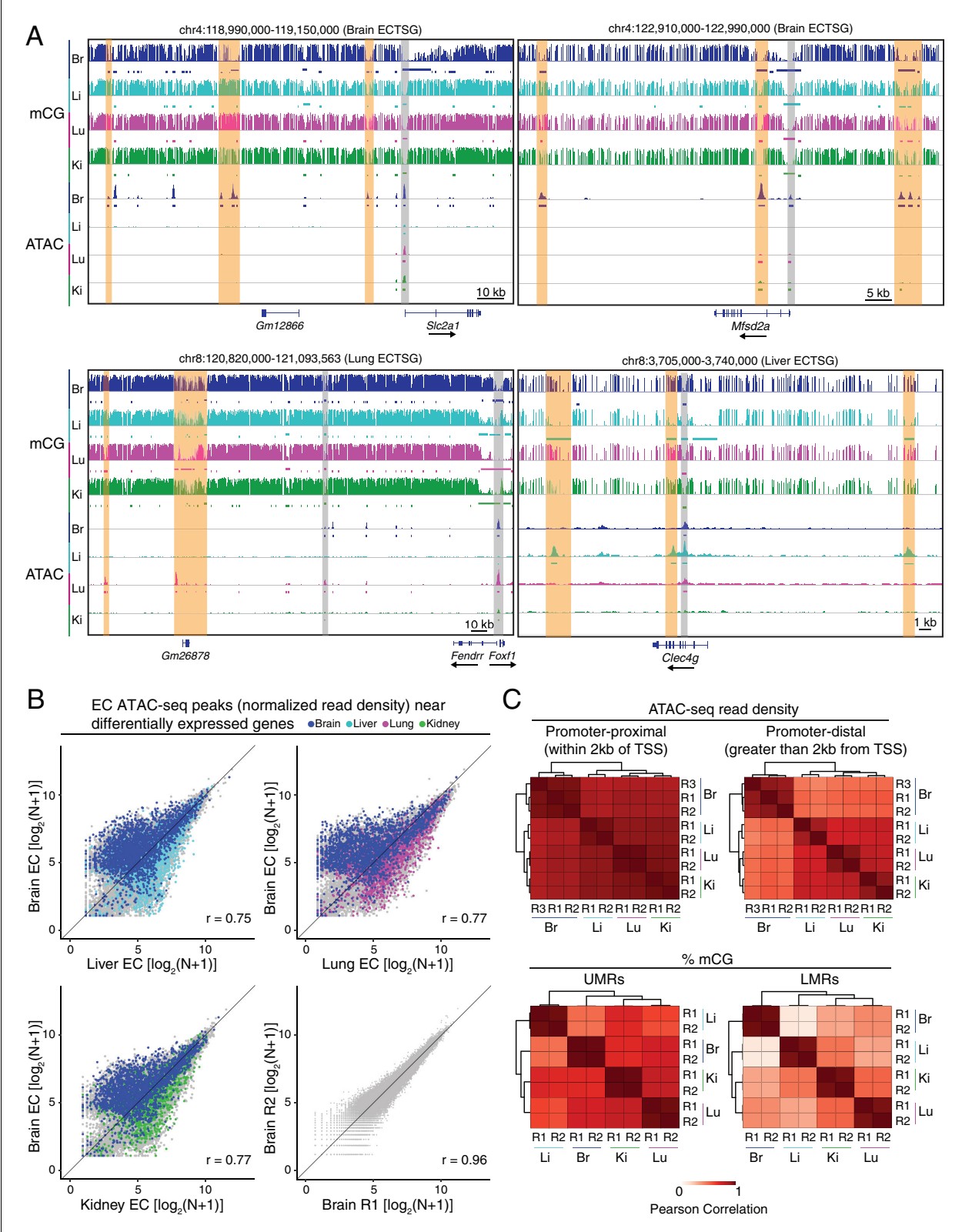

**Figure 4.** Accessible chromatin and hypomethylated regions reveal candidate EC regulatory elements. (**A**) Genome browser images showing CG methylation (top) and accessible chromatin (bottom) around ECTSGs. Colored bars under mCG tracks mark UMRs (upper row) and LMRs (lower row). For accessible chromatin, histograms of ATAC-seq reads are shown. Colored bars under ATAC tracks indicate the called ATAC-seq peaks. Vertical orange bars highlight co-localizing cell-type-specific open chromatin and differentially hypomethylated DNA. Vertical gray bars highlight shared regions

*Figure 4 continued on next page*

*Figure 4 continued*

of open chromatin in at least three EC subtypes. The ATAC-seq reads shown here represent the full range of ATAC-seq fragment sizes. (**B**) Scatter plots of normalized ATAC-seq read density (N) within ATAC-seq peaks called from <100 bp fragments. Values shown are $\log_2(N + 1)$. Colored symbols correspond to peaks for which the closest annotated TSS is a differentially expressed EC-enriched gene from the indicated tissue. Lower right, comparison between two brain EC ATAC-seq biological replicates. (**C**) Pairwise Pearson correlation heatmaps for ATAC-seq read density within ATAC-seq peaks at promoter-proximal (<2 kb from TSS; upper left) or promoter-distal (>2 kb from TSS; upper right) and percent CG methylation in UMRs (lower left) and LMRs (lower right).

DOI: https://doi.org/10.7554/eLife.36187.013

The following figure supplements are available for figure 4:

**Figure supplement 1.** Quality control for ATAC-seq.
DOI: https://doi.org/10.7554/eLife.36187.014

**Figure supplement 2.** EC subtype differences in distal epigenetic features.
DOI: https://doi.org/10.7554/eLife.36187.015

**Figure supplement 3.** Differentially accessible chromatin between peripheral ECs.
DOI: https://doi.org/10.7554/eLife.36187.016

**Figure supplement 4.** Relationship between accessible and hypomethylated regions of the EC genome.
DOI: https://doi.org/10.7554/eLife.36187.017

of UMRs overlap with ATAC-seq peaks in all four EC subtypes. In comparing ECTS-hypo-DMRs and ECTSAPs, <5% of liver EC, lung EC, and kidney EC hypo-DMRs overlap with their respective ATAC-seq peaks, whereas 13% of brain EC hypo-DMRs overlap with brain ATAC-seq peaks. As two explanations for this numerical difference, we suggest either (1) that many distal enhancer regions may not exhibit a sufficient degree of chromatin accessibility to permit insertion by two transposon complexes, the requisite event for detection of DNA accessibility by ATAC-seq, or (2) that distal enhancer regions may have greater variance in their occupancy across the population of cells, resulting in a general decrease in average accessibility.

## Tissue-specific EC transcription factors and their DNA targets

If LMRs and ATAC-seq peaks mark CREs, then they should be enriched for the DNA binding motifs of the TFs that control the distinct transcriptional, chromatin, and DNA methylation landscapes observed in ECs from different tissues. Current models of transcriptional regulation posit that TFs act in a combinatorial fashion, co-localizing at CREs to give rise to cell-type-specific transcription (*Heinz and Glass, 2011*; *Heinz et al., 2015*). These models predict that CREs in different EC subtypes should (1) share motifs corresponding to TFs that orchestrate patterns of gene expression common to all ECs, and (2) exhibit subtype-specific motifs corresponding to TFs that orchestrate patterns of gene expression distinctive to each EC subtype.

To investigate these models, we used Hypergeometric Optimization of Motif EnRichment (HOMER), a suite of tools for discovering enriched motifs in genomic sequences (*Heinz et al., 2010*). The HOMER de novo motif detection strategy parses genomic sequences under consideration into all possible k-mers of a desired motif length and searches for enrichment of each k-mer, a strategy that is not dependent on existing biochemical definitions of TF motifs (*Figure 5A*). The results of the k-mer analysis are then compared to a compendium of known TF motifs. HOMER also directly searches the genomic sequences under consideration for known TF motifs (*Figure 5B*). In general, we found the two approaches to be in good agreement.

To address the first prediction, we first identified LMRs and ATAC-seq peaks that are not present in photoreceptors and brain neurons (*Mo et al., 2015*; *Mo et al., 2016*). This should filter out CREs associated with 'housekeeping' genes, thereby enriching for CREs that reflect the EC regulatory landscape. This procedure identified 56,755 LMRs and 12,200 ATAC-seq peaks that are relatively EC-specific (*Supplementary files 4* and *6*). From this subset, 8035 LMRs and 899 ATAC-seq peaks were common to all four EC subtypes. Analyzing this common subset using HOMER revealed a highly significant (LMRs: p-value=$10^{-1691}$; ATAC-seq peaks: p-value=$10^{-148}$) enrichment over random GC-matched genomic regions for a sequence that resembles the binding motif of class 1 members of the ETS family of TFs, which includes ERG, FLI1, ETV2, and ETS1 (*Wei et al., 2010*; *Hollenhorst et al., 2011*); *Figure 5—figure supplement 1A*). The finding of a common enriched class 1 ETS family motif is not unexpected given the large body of data supporting a role for class 1

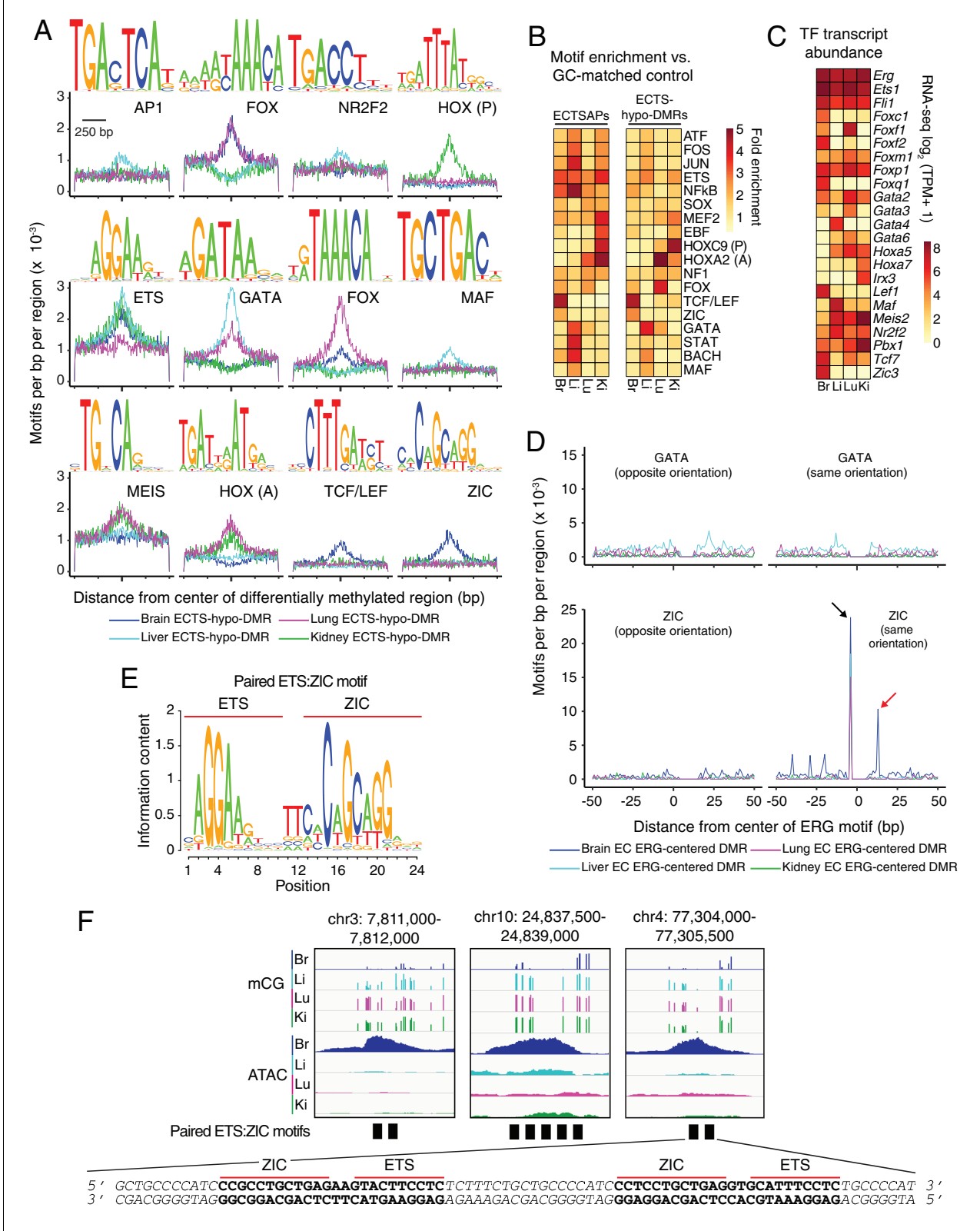

**Figure 5.** Motif enrichment analysis identifies candidate TF regulators of tissue-specific EC development and function. (**A**) HOMER-identified enriched motifs in ECTS-hypo-DMRs. Frequency of the indicated motif as a function of distance from the center of ECTS-hypo-DMRs. Shown above each individual plot is the position weight matrix (PWM) of the enriched nucleotide sequence. The TF family that most closely matches the motif is indicated below the PWM. (**B**) Heatmap showing the fold enrichment for the indicated TF motifs (% ECTSAPs or ECTS-hypo-DMRs containing the motif divided

*Figure 5 continued on next page*

*Figure 5 continued*

by % GC-matched background genomic regions containing the motif). Representative members of TF families that exhibited a significant enrichment (FDR < 0.001) are shown. (**C**) Heatmap showing TPMs for the four EC subtypes for a subset of TFs with the motifs shown in (**A**). Values shown are $log_2$(TPM +1). (**D**) ECTS-hypo-DMRs were centered on the motif for ERG, a member of the ETS family, and the frequencies of the indicated motifs were plotted as a function of distance from the ERG motif with a bin size of 1 bp. Black arrow: the ZIC motif ends with the sequence AGG and the ERG motif begins with the sequence AGG, thereby generating a frequency spike in all four EC subtypes at position −5 bp that represents the overlap of the two sites. Red arrow: the frequency spike for the ZIC motif at +11 bp is only present in one orientation and only in brain ECTS-hypo-DMRs centered on the ERG motif. (**E**) PWM for the consensus sequence of the paired ETS:ZIC motif. (**F**) Representative instances of the paired ETS:ZIC motif from brain ECTS-hypo-DMRs. Each instance is represented by a black rectangle. The bottom strand of the sequence in (**F**) matches the consensus sequence shown in (**E**).
DOI: https://doi.org/10.7554/eLife.36187.018

The following figure supplements are available for figure 5:

**Figure supplement 1.** Candidate TF regulators of EC gene expression.
DOI: https://doi.org/10.7554/eLife.36187.019
**Figure supplement 2.** TF binding motifs in candidate CREs near ECTSGs.
DOI: https://doi.org/10.7554/eLife.36187.020

ETS TFs in EC development and function (*McLaughlin et al., 2001*; *Pham et al., 2007*; *Meadows et al., 2011*; *Oh et al., 2015*; *Sumanas and Choi, 2016*). We also detected k-mers matching the SOX (p-value=$10^{-101}$) and GATA (p-value=$10^{-94}$) binding motifs in shared EC-only LMRs and - unexpectedly - a k-mer matching the telomere sequence TTAGGG (p-value=$10^{-123}$) in shared EC-only ATAC-seq peaks (*Figure 5—figure supplement 1A*).

To address the second prediction, we used HOMER to search for TF motifs in sequences present in the four sets of ECTSAPs and the four sets of ECTS-hypo-DMRs. In addition to the EC subtype-specific enriched motifs described in the paragraphs below, this analysis revealed the class 1 ETS motif in both ECTSAPs (p-value=$10^{-45}$–$10^{-718}$) and ECTS-hypo-DMRs (p-value=$10^{-30}$–$10^{-1169}$) (*Figure 5A and B*). This observation suggests that ETS family TFs not only act in shared CREs to determine and maintain common aspects of the EC phenotype, but they also cooperate with EC subtype-specific TFs in EC subtype-specific CREs. As determined from the EC RNA-seq datasets, all four subtypes of ECs express members of the ETS family of TFs, including ERG, FLI1, and ETS1 (*Figure 5C*). As seen in *Figure 5B*, the motif analysis using ECTSAPs was in good agreement with the motif analysis using ECTS-hypo-DMRs, but the latter exhibited a higher signal-to-noise ratio, most likely due to the larger number of ECTS-hypo-DMRs.

Significantly, HOMER identified several EC subtype-specific enriched motifs (*Figure 5A and B*). As different members of a given TF family generally recognize the same or nearly the same motif, we determined which family members are likely to be relevant for tissue-specific EC transcriptional regulation by assessing their transcript abundances in the EC RNA-seq datasets (*Figure 5C*). A summary of motifs and TFs for each of the four EC subtypes follows (see also *Supplementary file 7*).

- In brain ECTS-hypo-DMRs, we identified enriched k-mers that closely match the motifs of the Forkhead box (FOX), TCF/LEF, and ZIC TF families (*Figure 5A*). Transcripts corresponding to three *Fox* family members (*Foxq1*, *Foxc1*, and *Foxf2*), two *Tcf/Lef* family members (*Tcf7* and *Lef1*), and the *Zic* family member *Zic3* were highly enriched in brain ECs (*Figure 5C*). LEF/TCF family members are the effectors of canonical Wnt signaling in the nucleus (*Cadigan and Waterman, 2012*). The specific enrichment of LEF/TCF motifs in brain ECTS-hypo-DMRs and ECTSAPs is consistent with previous evidence that canonical Wnt signaling in ECs plays a central role in CNS angiogenesis and BBB development and maintenance, but that it has little or no role in vascular development outside of the CNS (*Liebner et al., 2008*; *Stenman et al., 2008*; *Daneman et al., 2009*; *Wang et al., 2012*; *Zhou et al., 2014*; *Cho et al., 2017*). The brain EC-specific enrichment of ZIC motifs is consistent with previous work showing that the accumulation of ZIC3 in the developing retinal vasculature depends on NORRIN/FRIZZLED4 (i.e. canonical Wnt) signaling (*Wang et al., 2012*). Analysis of candidate CREs near the *Zic3* gene in all four EC subtypes revealed the presence of a brain ECTSAP ~50 kb downstream of the gene that contains ETS and TCF/LEF binding sites, suggestive of a nearby Wnt-responsive enhancer element (translucent orange bar in *Figure 5—figure supplement 2A and B*).

- In liver ECTS-hypo-DMRs, we identified enriched k-mers that closely match the motifs recognized by members of the GATA and MAF TF families (*Figure 5A*). RNA-seq across the four EC

populations showed that *Gata4* and *Maf* have the greatest enrichment among members of their respective families in liver ECs (*Figure 5C*). Consistent with these observations, a recent study found that GATA4 controls liver EC development and function (*Géraud et al., 2017*). In liver ECTS-hypo-DMRs, we also detected enrichment for a k-mer that corresponds to the NR2F2/COUP-TFII motif (*Figure 5A*). As NR2F2/COUP-TFII is essential for establishing venous EC identity (*You et al., 2005*), enrichment of this motif could reflect enrichment of venous ECs due to inclusion of the portal vein and its tributaries or a bona fide role for NR2F2 in liver sinusoid development. Like *Gata4* and *Maf*, *Nr2f2* transcripts are enriched in liver ECs (*Figure 5C*).

 - In lung ECTS-hypo-DMRs, we identified enriched k-mers that closely match the motifs of the FOX, GATA, and homeobox TF families (*Figure 5A*). *Foxp1*, *Foxm1*, *Foxf1*, *Gata2*, *Gata3*, *Gata6*, and *Hoxa5* transcripts are enriched in lung ECs (*Figure 5C*).

 - In kidney ECTS-hypo-DMRs, we identified enriched k-mers that closely match the homeobox motif (*Figure 5A*). RNA-seq shows strong kidney-specific enrichment for homeobox transcripts *Irx3*, *Hoxa7*, *Pbx1*, and *Meis2* (*Figure 5C*). A motif corresponding to anterior HOX proteins (e.g. HOXA2; *Figure 5B*) was enriched in lung and kidney whereas a motif corresponding to posterior HOX proteins (e.g. HOXC9; *Figure 5B*) showed enrichment primarily in kidney, consistent with the *Hox* gene expression patterns described above (*Figure 3* and *Figure 3—figure supplement 1*).

If the enriched motifs play a role in regulating EC subtype-specific gene expression, then one might expect an enrichment for the same motifs in candidate EC subtype-specific CREs near ECTSGs in one EC subtype relative to candidate CREs near ECTSGs in other EC subtypes. Consistent with this expectation, ECTS-hypo-DMRs found within 100 kb of ECTSGs exhibit statistically significant enrichment for most of the motifs in *Figure 5A* (*Figure 5—figure supplement 1B*). ECTSAPs within 100 kb of ECTSGs were also significantly enriched for several of these motifs (*Figure 5—figure supplement 1C*). *Figure 5—figure supplement 2* shows five examples of individual ECTSGs and their nearby candidate CREs with tissue-specific TF and ETS motifs marked in the regions encompassed by ECTS-hypo-DMRs and ECTSAPs (*Figure 5—figure supplement 2C–F*; data are shown for Zic3 and for the four genes shown in *Figure 4A*).

## Identification of a paired ETS and ZIC motif with precise spacing and orientation

If ETS TFs, which are expressed in all ECs, cooperate with tissue-specific EC TFs to control expression of ECTSGs, then there might be a non-random spatial relationship between ETS motifs and motifs for tissue-specific TFs. To test this idea, we compiled all of the ECTS-hypo-DMRs containing an ERG (i.e. ETS family) motif, aligned these ECTS-hypo-DMRs on the ERG motif, and then generated, for each of the four EC subtypes, motif frequency histograms with a 1 bp bin size for the lung EC-enriched FOXF1 motif, the liver EC-enriched GATA4 motif, the kidney EC-enriched HOXC9 motif, the brain EC-enriched ZIC3 and TCF/LEF motifs as a function of distance from the center of the ERG motif (*Figure 5D*; *Figure 5—figure supplement 1D*).

From this analysis, two general patterns emerged. One pattern is exemplified by the frequency of GATA4 motifs in liver ECTS-hypo-DMRs, which are modestly enriched within ~40 bp of the ERG motif. This suggests that in liver ECs GATA4 may collaborate with ERG and/or other class 1 ETS factors but without forming a unique ETS-GATA4-DNA ternary complex. The second pattern is exemplified by the spatial relationship between the ERG and ZIC3 motifs. Here, we observed a spike in ZIC3 motif frequency precisely 2 bp from the end of and in the same orientation as the ERG motif only in brain ECTS-hypo-DMRs (red arrow in *Figure 5D* lower right panel). A partial overlap of the ERG and ZIC3 motifs (the nucleotide sequence AGG occurs at the start of the ERG motif and at the end of the ZIC3 motif) generates a second spike in ZIC3 motif frequency that is present in all four EC subtypes but is unlikely to be of biological significance (black arrow in *Figure 5D* lower right panel). Additional correlations are observed between the ERG motif and FOXF1 and HOXC9 motifs in lung and kidney ECTS-hypo-DMRs, respectively (red arrows in *Figure 5—figure supplement 1D*). The paired FOX:ETS motif in lung ECTS-hypo-DMRs matches a composite FOX:ETS motif found in several EC-specific enhancers that is known to be bound and synergistically activated by FOX and ETS factors (*De Val et al., 2008*; *Robinson et al., 2014*). The paired ETS:HOX motif in kidney ECTS-hypo-DMRs suggests a similar interaction between ETS factors and HOX factors; however, we found no previous characterization of such an interaction in the literature. A partial overlap of the nucleotide sequence AGG between the TCF/LEF motif and the ERG motif generates a spike in frequency

that is present in all four EC subtypes but is unlikely to be of biological significance (black arrow in *Figure 5—figure supplement 1D*).

The proximity, precise spacing, and orientation of the ERG and ZIC3 motifs in brain ECTS-hypo-DMRs suggests that these two transcription factors form a ternary complex by binding to this larger DNA element. To determine a consensus for this element, we repeated the HOMER analysis on brain ECTS-hypo-DMRs with longer motifs. This analysis revealed a consensus motif that we will refer to as the paired ETS:ZIC motif (*Figure 5E*; *Supplementary file 7*). Strikingly, identically spaced ETS and ZIC motifs lead to robust enhancer activity in the ascidian *Ciona*, implying an ancient and conserved role for the cooperation of ETS factors and ZIC factors in transcriptional regulation (*Farley et al., 2016*). Genome-wide, there are only 1944 instances of the paired ETS:ZIC motif, and this paired motif is significantly enriched in candidate brain EC CREs (both hypo-DMRs and ATAC-seq peaks) relative to random GC-matched genomic regions (DMR p-value=$10^{-45}$; ATAC peak p-value=$10^{-17}$) and to peripheral EC CREs (DMR p-value=$10^{-74}$; ATAC peak p-value=$10^{-7}$; *Figure 5F*; *Figure 5—figure supplement 1E*). Interestingly, 81% of the brain EC CREs that contain one or more paired ETS:ZIC motifs overlap the long terminal repeats (LTRs) of the mouse-specific endogenous retrovirus (ERV) family RLTR45/ERVB4_2. Transposable elements, especially LTR retrotransposons, provide a rich genomic substrate for the evolution of non-coding regulatory elements (*Emera and Wagner, 2012*; *Chuong et al., 2013*; *Chuong et al., 2016*); reviewed in *Thompson et al., 2016*). We speculate that RLTR45/ERVB4_2 may represent an example of a transposable element family that is being co-opted into host gene regulatory networks that utilize ETS and ZIC factors.

## Canonical Wnt signaling in CNS ECs versus non-CNS ECs

At present, it is not known whether CNS EC-specific effects of canonical Wnt signaling reflect a spatial restriction on Wnt signaling - perhaps reflecting the spatial distribution of the relevant ligands - or whether canonical Wnt signaling occurs throughout the vasculature but has different downstream consequences in CNS and non-CNS tissues. To address this question, we visualized canonical Wnt signaling at cellular resolution in CNS and non-CNS vasculature using an EC-specific Cre recombinase (*Tie2-Cre*) and a Cre-dependent Wnt reporter (*R26-Tcf/Lef-LSL-H2B-GFP-6xMYC*). With this combination of genetic elements, Cre-mediated excision of a *loxP*-flanked transcription stop cassette allows the multimerized LEF/TCF motifs, together with a minimal promoter, to report canonical Wnt signaling by controlling production of a nuclear-localized histone H2B-GFP-6xMYC fusion protein specifically in ECs (*Cho et al., 2017*).

In coronal sections of E13.5 embryos, the pan-EC TF ERG and the pan-EC plasma membrane protein ICAM2 mark all ECs (*Figure 6A and A'*). At this stage, BBB development is already underway, as judged by the accumulation of the glucose transporter GLUT1/SLC2A1, a BBB marker, specifically in CNS and perineural ECs (*Figure 6B and B'*). At E13.5, ZIC3 accumulation in the vasculature is also limited to CNS and perineural ECs (*Figure 6C and C'*; *Figure 6—figure supplement 1*), a distribution that closely matches that of GLUT1/SLC2A1. Outside of the vasculature, ZIC3 accumulates in developing neurons in the ventral CNS. Similarly, vascular accumulation of LEF1, which is both a mediator and a marker of canonical Wnt signaling, is limited to CNS and perineural ECs (*Figure 6—figure supplement 2A and A'*). Visualizing the accumulation of the Wnt reporter, H2B-GFP-6xMYC, with anti-MYC immunostaining shows a neural and perineural distribution, closely matching that of SLC2A1, ZIC3, and LEF1 (*Figure 6*; *Figure 6—figure supplement 1*). We note that cells positive for the Wnt reporter occur outside of the CNS, yet their rounded morphology and lack of association with blood vessels suggests that these cells are macrophages or other immune cells, consistent with the specificity of *Tie2-Cre*. At P7, H2B-GFP-6xMYC accumulates in CNS ECs, as well as ECs in the renal medulla (*Figure 6—figure supplement 2B–E*), and ZIC3 and SLC2A1 are restricted to the CNS vasculature (data not shown). These protein distributions are consistent with the CNS EC-specific expression of *Lef1* and *Zic3* found by RNA-seq (*Figure 5C*) and the CNS EC-enrichment of TCF/LEF and ZIC3 motifs in ECTS-hypo-DMRs (*Figure 5A and B*). Taken together, these data imply that, among ECs, canonical Wnt signaling is largely confined to the CNS, with a corresponding restriction in the expression of TFs that mediate (LEF1) and respond to (LEF1 and ZIC3) canonical Wnt signaling.

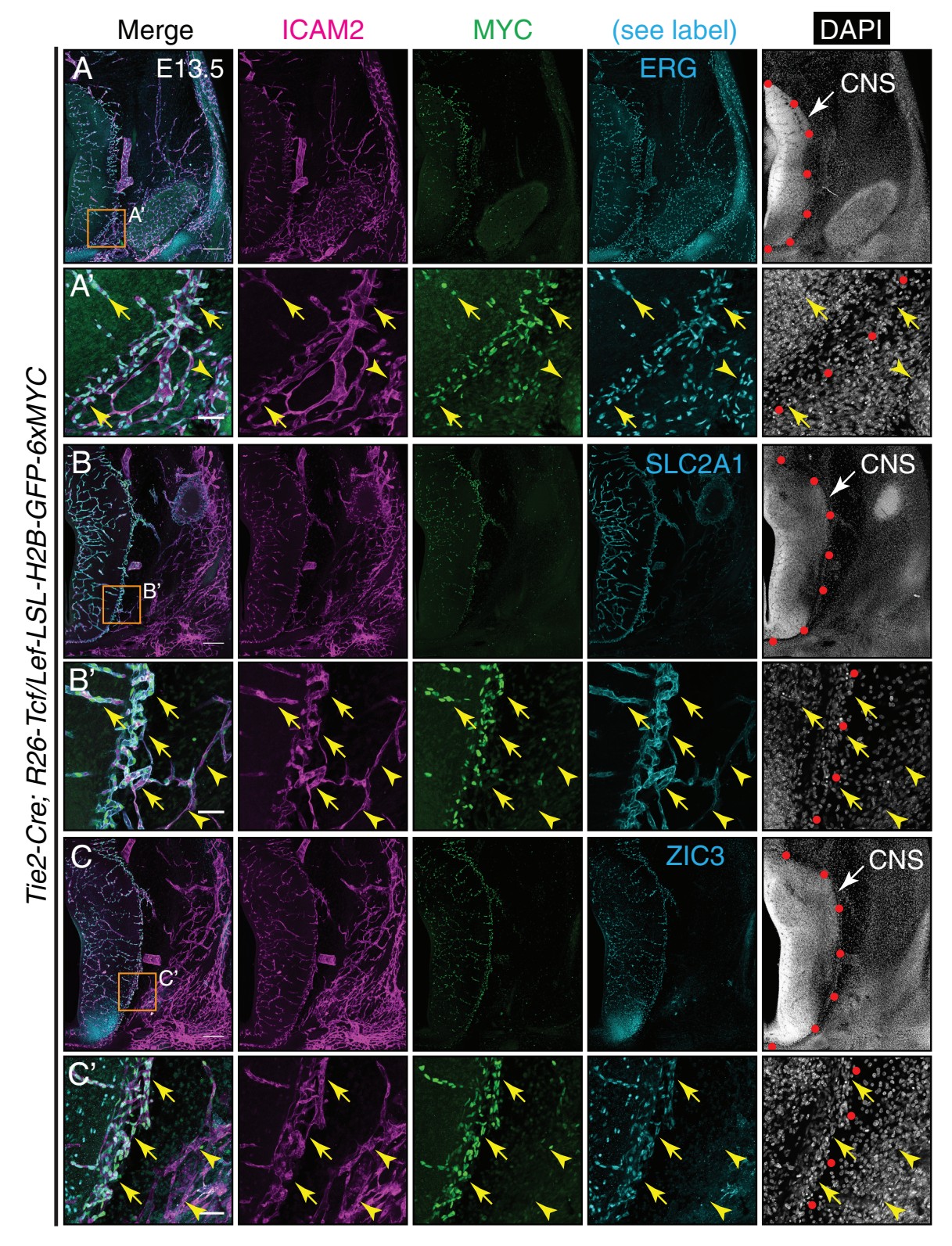

**Figure 6.** Canonical Wnt signaling in CNS but not peripheral ECs at E13.5. (**A–C**) Coronal sections of E13.5 *Tie2-Cre;R26-Tcf/Lef-LSL-H2B-GFP-6xMYC* embryos near the cephalic flexure. The markers are: ICAM2 (pan-EC membrane protein), MYC (the canonical Wnt reporter), ERG (pan-EC TF), SLC2A1 (the glucose transporter GLUT1; a BBB marker), ZIC3, and DAPI. (**A'–C'**) Higher magnification of the boxed regions in (**A–C**). The boundary between CNS and peripheral tissue is marked on the DAPI image with red circles. The nuclear MYC signal reveals canonical Wnt signaling in CNS ECs (yellow

*Figure 6 continued on next page*

Figure 6 continued

arrows) but not in peripheral ECs (yellow arrowheads). ZIC3 is present in CNS ECs (yellow arrows) but not in peripheral ECs (yellow arrowheads), and in developing neurons in the ventral CNS. Scale bars in A, B, and C: 200 um. Scale bars in A', B', and C': 50 um.

DOI: https://doi.org/10.7554/eLife.36187.021

The following figure supplements are available for figure 6:

**Figure supplement 1.** Quantification of immunostaining in *Figure 6*.

DOI: https://doi.org/10.7554/eLife.36187.022

**Figure supplement 2.** LEF1 in ECs in the E13.5 CNS, and canonical Wnt signaling in the P7 brain, liver, lung, and kidney.

DOI: https://doi.org/10.7554/eLife.36187.023

## Single-cell RNA-seq reveals heterogeneity among brain ECs

Up to this point, the focus has been on EC heterogeneity between tissues. Here, we address the question of intra-tissue EC heterogeneity. It is well established that arterial, venous, and capillary ECs represent distinct subtypes based on morphology, function, and gene expression profiles (*Aird, 2007a*; *Potente and Mäkinen, 2017*), but it is not clear whether or to what extent these EC subtypes can be further subdivided. It is also not clear how developing EC subtypes, such as tip cells and proliferating cells, are related to the more mature EC subtypes. To address these questions, we assessed EC gene expression in the developing CNS at single-cell resolution by performing single-cell RNA-seq (scRNA-seq) on 3,946 FACS-purified GFP-positive ECs from a P7 *Tie2-GFP* mouse brain.

scRNA-Seq data were processed and analyzed using a modified dpFeature approach as described in the Monocle R/Bioconductor package (*Trapnell et al., 2014*). t-Distributed Stochastic Neighbor Embedding (t-SNE) of cells revealed a single interconnected distribution of gene expression profiles suggesting a continuity of transcriptional identity across brain ECs at this developmental time point consistent with the observed zonation of EC subtypes along the vasculature. Subsequent clustering analysis identified six EC clusters (*Figure 7A*) that could be assigned as tip, mitotic, venous, or arterial cells (one cluster each) or capillary cells (two adjacent clusters) based on known EC subtype marker gene expression (*Figure 7B*). One of the capillary clusters is enriched for cells expressing *DOPA Decarboxylase* (*Ddc*), a gene known to be expressed in ECs in bovine aorta (*Sorriento et al., 2012*). The ECs in the high-*Ddc* cluster also express multiple genes that are characteristic of arterial ECs and this cluster was therefore named 'capillary-A' (capillary-arterial). The second capillary cluster was relatively deficient in *Ddc*-expressing cells, and it was enriched for cells that express genes characteristic of venous ECs. The second cluster was therefore named 'capillary-V' (capillary-venous). Mitotic ECs express multiple genes that are also expressed in capillary-V and venous ECs (*Figure 7C*), which is consistent with the observation that proliferating cells arise from veins during sprouting angiogenesis (*Ehling et al., 2013*; *Xu et al., 2014*; *Hasan et al., 2017*; *Pitulescu et al., 2017*). Capillary-A ECs express multiple genes that are also expressed in tip cells and arterial ECs, but tip cells and arterial ECs show less mutual similarity (*Figure 7C*).

The distribution of DDC was examined by immunostaining of P7 WT mouse brain (*Figure 7—figure supplement 1*). DDC was localized to puncta, most likely corresponding to axon terminals of dopaminergic neurons; this pattern was particularly concentrated in the striatum, a brain region receiving dense dopaminergic innervation (*Figure 7—figure supplement 1C and D*). Most interestingly, we detected both fine- and coarse-grained heterogeneity in DDC immunostaining in GS Lectin-positive capillaries in different anatomic structures within the same brain sections, as seen by comparing the intensity of the general EC marker GS-Lectin and anti-DDC immunoreactivity. Larger vessels showed minimal DDC immunoreactivity (yellow arrows in *Figure 7—figure supplement 1C*). The heterogeneity in DDC immunoreactivity among capillary ECs likely corresponds to the differential capillary-A versus capillary-V gene expression patterns defined by scRNA-seq.

Although ECs within the six clusters express many genes in common, there were multiple genes expressed more or less selectively by each cluster (*Figure 7C*; *Supplementary file 8* lists the top 25 genes for each cluster). The identification of multiple new markers for tip cells is especially interesting as it substantially extends earlier screens for tip cell markers based on the overproduction of tip cells in Dll4$^{+/-}$ mouse retinas and on laser capture microdissection (*del Toro et al., 2010*; *Strasser et al., 2010*). Five genes that were previously validated as tip cell markers in the retina —

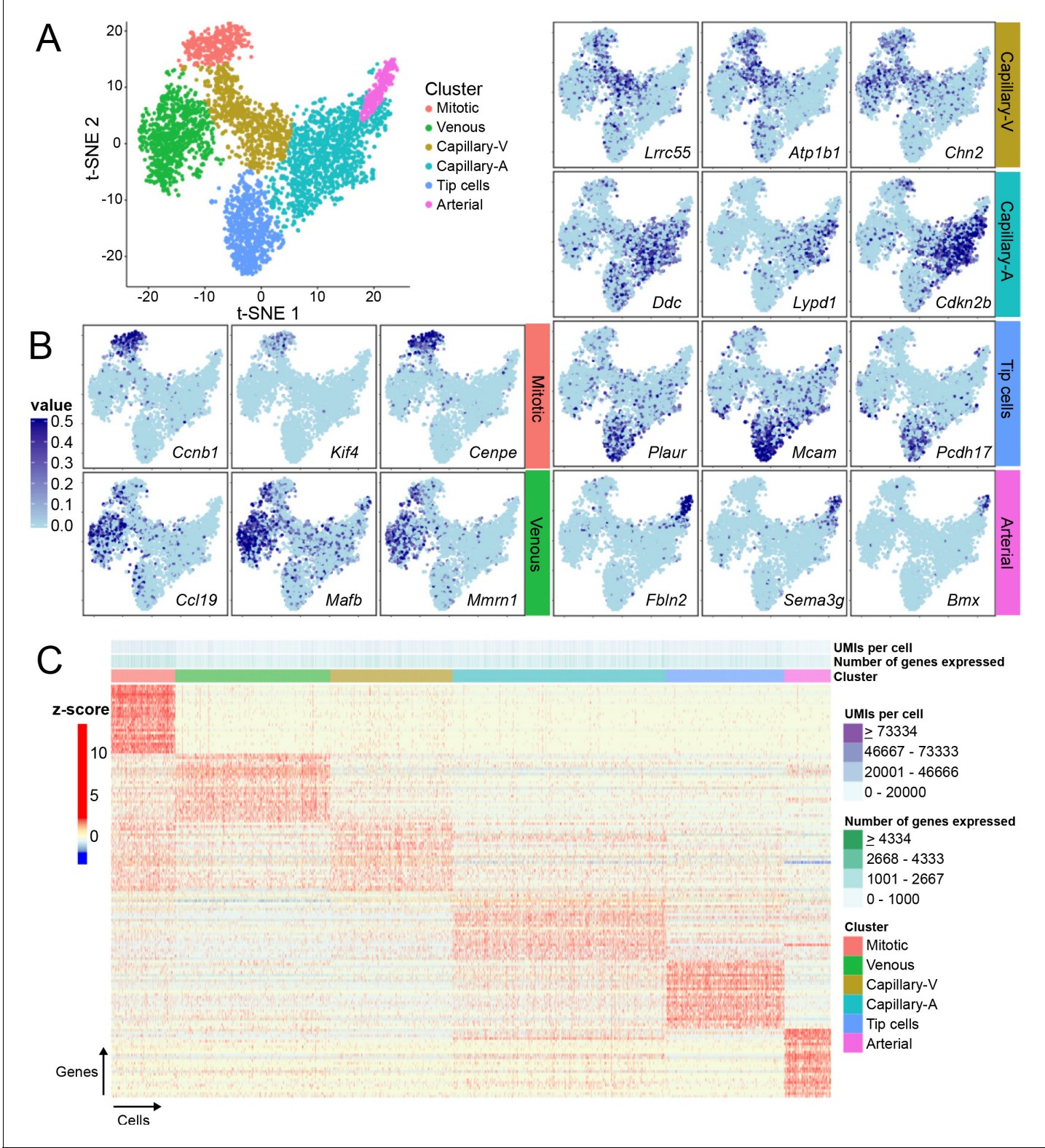

**Figure 7.** Single-cell RNA-seq of P7 brain ECs reveals intra-tissue heterogeneity. (**A**) t-SNE plot of 3946 P7 brain ECs showing six clusters corresponding to tip cells, and mitotic, venous, capillary-venous (Capillary-V), capillary-arterial (Capillary-A), and arterial ECs. (**B**) The t-SNE plot from (**A**) showing expression of three marker genes with enriched expression for each EC cluster. Cells with no RNA-seq reads are shown in light blue; darker blue represents greater number of reads. (**C**) Heatmap showing scaled expression (z-scores) for the 25 most enriched marker genes for each EC cluster. *Supplementary file 8* lists the genes plotted in (**C**). Rows represent genes and columns represent cells.

*Figure 7 continued on next page*

*Figure 7 continued*

DOI: https://doi.org/10.7554/eLife.36187.024

The following figure supplement is available for figure 7:

**Figure supplement 1.** DOPA decarboxylase is expressed non-uniformly in the CNS vasculature.

DOI: https://doi.org/10.7554/eLife.36187.025

*Plaur*, *Angpt2*, *Lcp2*, *Cxcr4*, *Apln,* and *Kcne3* — are enriched in the scRNAseq tip cell cluster (*del Toro et al., 2010*; *Strasser et al., 2010*; *Rattner et al., 2013*). Interestingly, 19 of the 25 most enriched tip cell transcripts from the scRNA-seq analysis code for membrane or secreted proteins, many of which are known or suspected regulators of information flow. These include ion channels (the mechanosensor PIEZO2, voltage-gated sodium channel beta subunit SCN1B, and potassium channels KCNE3 and KCNA5), cell adhesion proteins (protocadherin PCDH17, cell adhesion molecules MADCAM1 and MCAM, and lectin CLEC1A), receptor tyrosine kinase signaling regulator SIRPA, G-protein-coupled receptor CMKLR1, plasminogen regulator SERPINE1/PAI, urokinase receptor PLAUR, ligands adrenomedullin (ADM) and ANGPT2, and ECM protein SMOC2. Among the six intracellular proteins in the list of top 25 tip cell markers, most are involved in signaling, including cyclic nucleotide phosphodiesterase 4B (PDE4B), serine/threonine phosphatase PPM1J/PP2C-zeta, NADPH oxidase organizer-1 (NOXO1), and the ubiquitin ligase HECW2, which regulates EC-EC junction stability via its substrate angiomotin-like-1 (*Choi et al., 2016*).

To independently assess the expression of candidate tip cell genes, we performed whole mount in situ hybridization (ISH) on P5-7 retina (*Figure 8*). At this developmental stage, the vascular plexus on the vitreal face of the retina is growing outward from the optic disc and tip cells are localized to the growing vascular front. Consistent with the scRNA-seq analysis, *Apln*, *Mcam*, *Lamb1*, and *Trp53i11* exhibited enriched expression at the angiogenic front of the superficial vascular plexus (*Figure 8A–D*), indicating tip cell enrichment of these genes. As a specificity control, *Tm4sf1* exhibited only arterial expression by both scRNA-seq analysis and whole mount retina ISH (*Figure 8E*).

To describe the relationships between the six EC clusters, the scRNA-seq data were used to construct a cell trajectory map (*Figure 9A*). This analysis produced a map with two branch points and four terminal states [mitotic (M), venous (V), arterial (A), and tip cell (T)]. Capillary-V and capillary-A ECs populate the regions adjacent to the first and second branch points, respectively. The first branch point represents a choice between the venous state and the arterial and tip-cell states, implying that, at the transcriptome level, tip cells are more closely related to arterial ECs than to venous ECs. As examples of gene expression patterns that support this model, *Hey1* is more highly expressed in arterial ECs and tip cells than in venous ECs, whereas *Nr2f2* is more highly expressed in mitotic and venous ECs than in arterial ECs or tip-cells (*Figure 9B*). The second branch point represents a choice between the arterial state and the tip-cell state. *Fbln2* and *Plaur* are examples of genes that are more highly expressed in arterial ECs and tip-cells, respectively (*Figure 9B*).

Among the transcripts with intriguing differences between EC clusters were several coding for cyclin-dependent kinase (CDK) inhibitors. In particular, *Cdkn1c/p57* expression was enriched in artery and vein ECs; *Cdkn1a/p21* expression was enriched in mitotic cells, tip cells, and capillary-A ECs; and *Cdkn2b/p15* expression was enriched in capillary-A ECs (*Figure 9—figure supplement 1A*). *Cdkn1a/p21* expression suggests that these ECs have exited the cell cycle (*Spencer et al., 2013*). Interestingly, TGF-beta2 (*Tgfb2*) was widely expressed but Latent TGF-beta-binding protein 4 (*Ltbp4*) expression was concentrated in arterial ECs, which utilize TGF-beta signaling to coordinate vascular smooth muscle cell development (*ten Dijke and Arthur, 2007*; *Figure 9—figure supplement 1A*). Transcripts coding for SMAD6 and SMAD7, two inhibitory SMADs induced by TGF-beta signaling (*Afrakhte et al., 1998*), were enriched in the capillary-A and arterial EC clusters (*Figure 9—figure supplement 1A*). Both CDKN1A and CDKN2B are known to mediate TGF-beta-induced cell cycle arrest (*Hannon and Beach, 1994*; *Datto et al., 1995*), consistent with a role for TGF-beta signaling in CNS EC development.

We also observed enrichment of transcripts coding for the chemokine receptor CXCR4 in tip cell, capillary-A, and arterial EC clusters, consistent with the role of this receptor in mediating angiogenic sprouting and artery formation (*Strasser et al., 2010*; *Bussmann et al., 2011*; *Ehling et al., 2013*; *Xu et al., 2014*; *Hasan et al., 2017*; *Pitulescu et al., 2017*); *Figure 9—figure supplement 1A*).

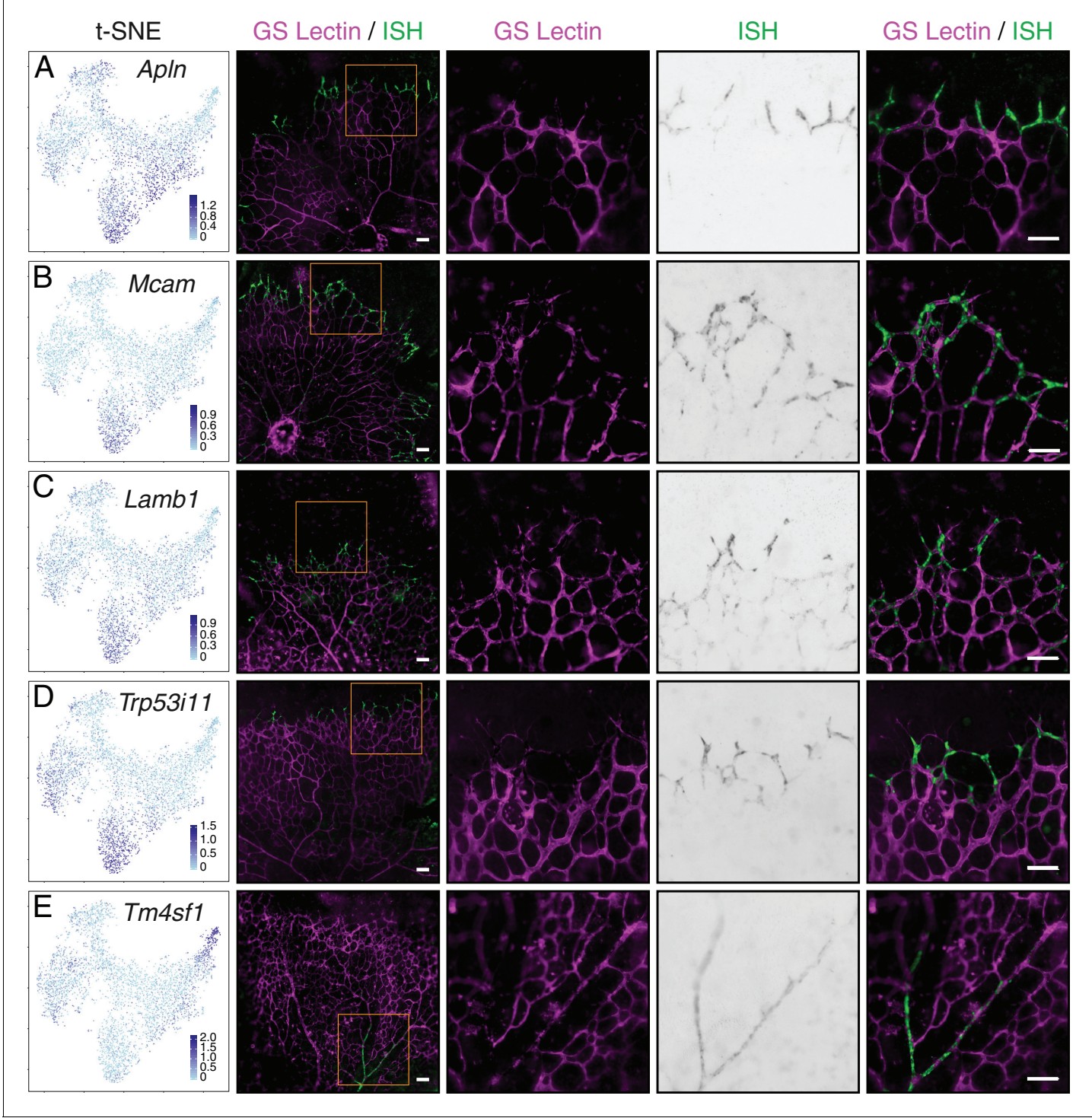

**Figure 8.** Single-cell RNA-seq of P7 brain ECs identifies novel tip cell markers. (A–E) Whole mount retina in situ hybridization (ISH) for known tip cell marker *Apln* (A); novel tip cell markers *Mcam* (B), *Lamb1* (C), and *Trp53i11* (D); and a novel arterial marker *Tm4sf1* (E). The first two columns, from left to right, are the t-SNE plot from *Figure 7A* showing expression of each marker gene and a low-magnification merged image of flatmount retina with blood vessels marked by GS Lectin (magenta) and ISH signal (green). The third to fifth columns showed the boxed region in column two at higher magnification, with separate signals (columns three and four) and the merged signals (column five). All scale bars are 50 um.
DOI: https://doi.org/10.7554/eLife.36187.026

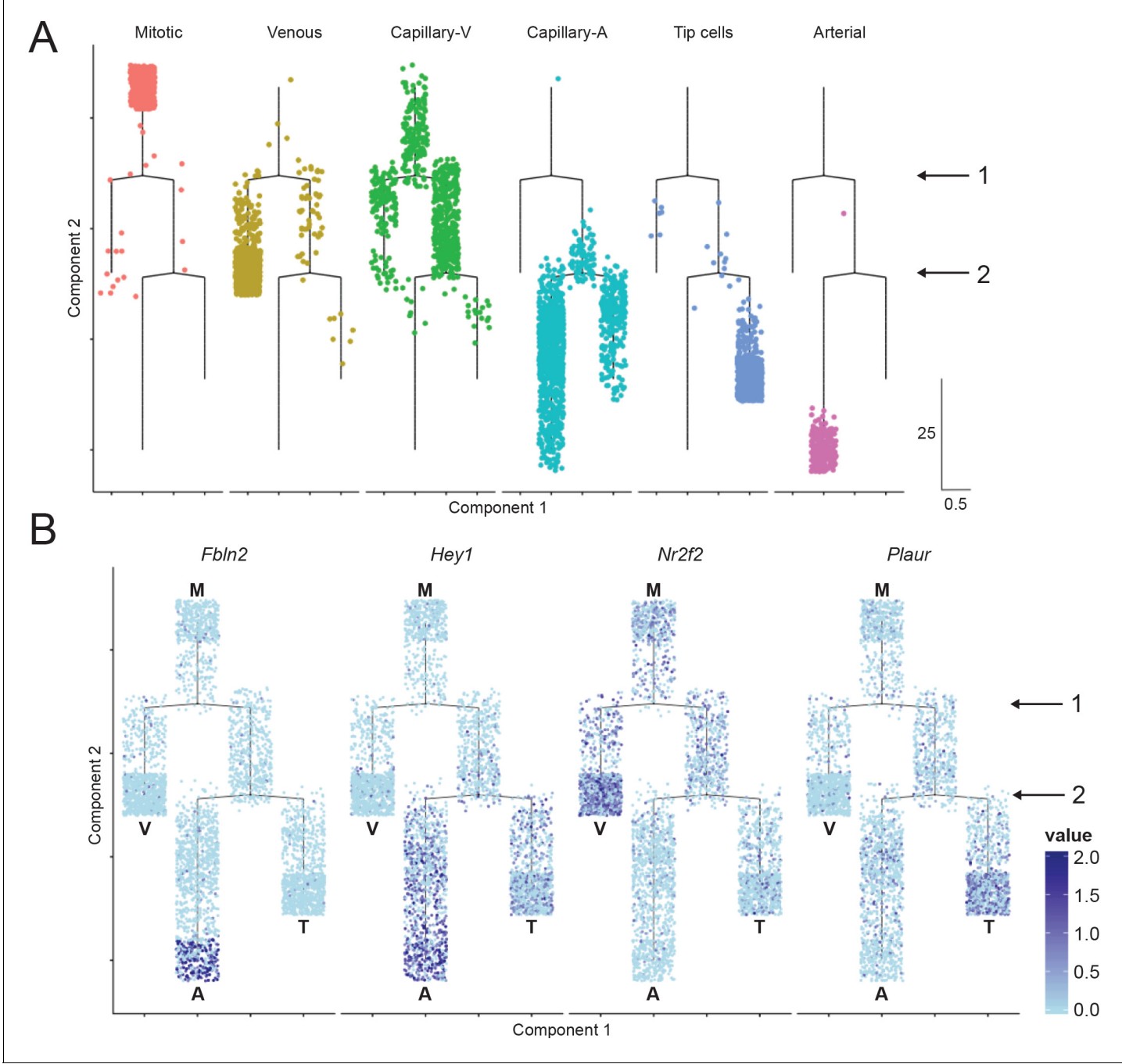

**Figure 9.** Cell trajectory analysis of brain ECs based on single-cell RNA-seq. (**A**) Plot showing the position of cells in each EC cluster on the constructed cell trajectory. (**B**) Summary of the two branch points (labeled '1' and '2') in the cell trajectory analysis. M, mitotic; V, venous; A, arterial; T, tip cells. Examples of markers that show differential expression as cells diverge from branch points 1 and 2. Cells with no RNA-seq reads are shown in light blue; dark blue represents greater number of reads. Hey1 is enriched in arterial and tip cells (i.e. the products of the right side of branchpoint 1); Nr2f2 is enriched in venous ECs; Fbln2 is enriched in arterial ECs; and Plaur is enriched in tip cells.

DOI: https://doi.org/10.7554/eLife.36187.027

The following figure supplement is available for figure 9:

**Figure supplement 1.** Brain ECs exhibit heterogeneous expression of cyclin-dependent kinase inhibitors, components of TGF-beta signaling, and components of CXCR4 signaling.

DOI: https://doi.org/10.7554/eLife.36187.028

Transcripts coding for CXCL12, the ligand for CXCR4, were highly enriched in the arterial EC cluster, suggesting that tip cells in the developing brain vasculature may receive a CXCL12 signal from arterial ECs. CXCL12 is expressed in pulmonary and coronary arterial ECs, where it mediates patterning of arterial vasculature (*Chang et al., 2017*; *Kim et al., 2017*). Cell trajectory representations of CDK inhibitor, TGF-beta signaling, and CXCR4/CXCL12 transcripts are shown in *Figure 9—figure supplement 1B*.

## Discussion

The experiments and analyses presented here define the transcriptome, accessible chromatin, and DNA methylome landscapes for ECs from brain, liver, lung, and kidney, and they relate differences in those landscapes to the regulatory programs that control tissue-specific EC heterogeneity. The resulting genome-wide analysis of candidate CREs, TF motifs, and TF expression reveal the distinctive gene regulatory architecture of each EC subtype. Using scRNA-seq, we define the relationships among developing and mature EC subtypes within the brain based on an unbiased clustering of gene expression profiles. In comparing CNS versus non-CNS ECs, the most prominent differences in TF motifs and TF expression include CNS EC-specific utilization of TCF/LEF and ZIC3 motifs and CNS EC-specific expression of TCF/LEF factors and ZIC3, implying a prominent role for canonical Wnt signaling and ZIC3 DNA binding in CNS EC-specific gene expression.

### The DNA methylation landscape as a guide to current and past gene expression

The relationship between patterns of mammalian DNA methylation and gene expression has been an object of inquiry for more than 40 years (*Riggs, 1975*; *Razin and Riggs, 1980*; *Ambrosi et al., 2017*). The discovery of diverse methyl CG binding proteins (*Shimbo and Wade, 2016*) and the more recent characterization of DNA methylation effects on the binding affinities of diverse TFs (*Hu et al., 2013*; *Kribelbauer et al., 2017*; *Yin et al., 2017*) imply that DNA methylation/demethylation can play a causal role in the control of chromatin state and the utilization of enhancers and promoters via its effects on protein binding. Reciprocally, changes in chromatin state and gene transcription can alter the accessibility of DNA to the enzymatic machinery of methylation and demethylation, leading to changes in DNA methylation (*Stroud et al., 2017*).

Comparative analyses of DNA methylation are especially powerful because technical developments now permit the genome-wide determination of DNA methylation at single-base resolution starting with small numbers of cells. Using this approach, we observe EC subtype-specific patterns of CG methylation that correlate with current patterns of gene expression and that also appear to reflect patterns of gene expression at earlier developmental points, especially at TF genes. As recognized in several previous studies, DNA methylation and gene expression do not exhibit a simple monotonic relationship (*Mo et al., 2015*; *Amabile et al., 2016*; *Liu et al., 2016*; *Neri et al., 2017*). Many expressed genes exhibit promoter demethylation, with a variable degree of demethylation extending into the gene body. Demethylation can also reflect earlier epochs in which the gene was expressed, thus serving as a mnemonic epigenetic feature (*Hon et al., 2013*; *Mo et al., 2015*). Paradoxically, in the HOX cluster, methylation appears to correlate inversely with current gene expression: it is lowest among CNS ECs, which show little or no HOX gene expression, and it is highest among kidney ECs, which show the highest levels of HOX gene expression. This inverse correlation could reflect a loss (or retention) of the association of Polycomb repressive complexes with HOX clusters (*Eskeland et al., 2010*; *Kundu et al., 2017*; *Lau et al., 2017*), a resulting loss (or retention) of chromatin compaction (*Chambeyron et al., 2005*; *Fraser et al., 2009*; *Ferraiuolo et al., 2010*; *Noordermeer et al., 2011*), and a corresponding increase (or decrease) in access to DNA methyltransferases.

Previous work has linked DNA demethylation of promoter-distal regions with enhancers (*Stadler et al., 2011*; *Schultz et al., 2015*). In the present study, LMRs and ECTS-hypo-DMRs have been used to identify candidate CREs and enriched TF motifs. As many promoter-distal hypomethylated regions are differentially hypomethylated in one or more pairwise comparisons between EC subtypes, the availability of the complete DNA methylomes for brain, liver, lung, and kidney ECs provides a powerful resource for identifying candidate tissue-specific EC CREs. While promoter-distal ATAC-seq peaks generally co-localize with LMRs and/or ECTS-hypo-DMRs, promoter-distal

hypomethylated regions are ~3 fold more numerous than promoter-distal regions of chromatin accessibility, and, as a result, TF motif enrichment analyses using hypomethylated regions provided greater statistical significance than the comparable analysis with accessible chromatin.

Rod photoreceptors exhibit a similar discrepancy in the number of promoter-distal hypomethylated features and accessible chromatin, and *Mo et al., 2016* suggested that the degree of chromatin compaction associated with the extremely small nuclei of rods could restrict access of DNA methyltransferases to DNA. As they mature, EC nuclei flatten and elongate (*Cao et al., 1998*; *Merks et al., 2006*; *Parsa et al., 2011*), which enhances chromatin compaction (*Versaevel et al., 2012*). Hence, it is not inconceivable that one of the explanations for the EC discordance between hypomethylated features and accessible chromatin may be EC nuclear structure.

## Organ-specific regulatory circuits for EC gene expression

Current evidence suggests that all ECs differentiate from hemangioblasts, a common precursor to both the vascular EC and hematopoietic lineages. Restricted expression of lineage determining TFs directs a population of mesodermal cells away from the blood cell fate and toward the EC fate, giving rise to EC progenitors that express the vascular endothelial growth factor (VEGF) receptor KDR (*Marcelo et al., 2013*; *Park et al., 2013*). In particular, the ETS factor ETV2 and the forkhead box factor FOXC2 cooperate to establish KDR +hemangioblasts (*De Val et al., 2008*; *Sumanas and Lin, 2006*; *Sumanas et al., 2008*; *Park et al., 2013*). Continued expression of ETV2 and other ETS factors, including ERG, FLI1, and ETS1, together with GATA2 and FOXC2, commits these progenitors to the EC lineage (*Pimanda et al., 2007*; *Liu and Patient, 2008*).

Arterial-venous differentiation represents the earliest observable differentiation of blood vessels in embryonic development. This process occurs through extracellular cues that activate specific transcriptional regulatory networks (reviewed in *Potente and Mäkinen, 2017*). In particular, VEGF and Notch signaling drive arterial differentiation and gene expression through SOX factors (*Fischer et al., 2004*; *Corada et al., 2013*; *Hermkens et al., 2015*), whereas NR2F2-mediated suppression of Notch signaling establishes venous endothelium (*You et al., 2005*). Although these more global EC transcriptional networks are well established, there remains a large gap in our understanding regarding the signaling pathways and transcriptional regulators that drive organ-specific EC identities.

In the present study, we provide transcriptional and epigenomic evidence for TFs that likely control tissue-specific specialization of ECs. For example, liver sinusoidal ECs exhibit differential expression of *Gata4* and *Maf* with a corresponding enrichment of GATA and MAF binding motifs in candidate CREs. Liver EC-specific deletion of *Gata4* leads to a conversion of sinusoidal endothelium to continuous endothelium similar to that found in brain vasculature (*Géraud et al., 2017*), and deletion of *Maf* globally results in a defective erythropoietic microenvironment in the liver (*Kusakabe et al., 2011*). Kidney ECs appear to utilize transcriptional networks that depend on homeobox TFs as seen by the enrichment of a HOX motif in kidney EC-specific CREs. Our observation that kidney ECs differentially express *Hoxa7*, *Pbx1*, and *Meis2* is consistent with a role for PBX and MEIS proteins as cofactors for HOX proteins (*Choe et al., 2014*; *Longobardi et al., 2014*; *De Kumar et al., 2017*). PBX1 is also essential in the developing renal mesenchyme (*Schnabel et al., 2003*), and its deletion in renal vascular mural cells leads to disrupted renal vascular development (*Hurtado et al., 2015*). The data presented here imply that PBX1 also plays a role in kidney ECs. Similarly, *Irx3* - another kidney EC-specific homeobox TF - was previously identified as a regulator of nephron segment identity in *Xenopus* (*Reggiani et al., 2007*), and the data presented here imply that it also plays a role in kidney ECs. These observations suggest that shared expression of TFs in parenchymal cells and ECs in the same organ may be a not uncommon pattern.

## Inter- and intra-tissue heterogeneity of vascular endothelial cells

The inter-tissue heterogeneity of EC structure and function has long been recognized as a central component of organ specialization (*Aird, 2007a*; *Aird, 2007b*). For example, in secondary lymphoid organs, leukocyte trafficking from the intravascular compartment to the surrounding parenchyma is mediated by leukocyte adhesion molecules on the luminal face of ECs in high endothelial venules (HEVs); in the kidney, serum filtration is controlled by ECs in renal glomeruli with a high density of fenestrations and a luminal glycocalyx that functions as a charge filter; in the liver, efficient

detoxification of xenobiotics requires rapid equilibration of serum contents with hepatocytes, a process that is facilitated by ECs with large trans-cellular fenestrae; and in the lung, regulation of systemic blood pressure is controlled by EC expression of angiotensin-converting enzyme (*Bakhle and Vane, 1974*).

For many tissues, culturing the resident ECs leads to a loss of tissue-specific EC markers, implicating paracrine signaling from the parenchyma in tissue-specific EC differentiation and maintenance. For example, HEV ECs isolated from human tonsils lose multiple tissue-specific markers after 2 days in culture (*Lacorre et al., 2004*). In some systems, co-culture with parenchymal cells leads to retention of organ-specific markers: co-culture of cardiac ECs with cardiac myocytes induces expression of markers characteristic of cardiac microvascular ECs (*Aird et al., 1997*), and co-culture of hepatic sinusoidal ECs with hepatocytes induces expression of markers characteristic of hepatic ECs (*Edwards et al., 2005*). With the exception of CNS ECs (discussed below), the identities of the organ-specific paracrine signals remain unknown.

The present genomic approach complements these foundational observations. Analyses of the transcriptome, accessible chromatin, and DNA methylome landscapes can be used to more precisely define organ-specific EC differentiation in vivo and in vitro, providing genomic snapshots of the different stages of EC differentiation, the effects of tissue-specific paracrine signals, and the response of ECs to disease-associated perturbations. With respect to the last point, this approach should prove especially interesting in dissecting the chromatin and epigenomic architecture of ECs within tumor vasculature, where a distinctive transcriptome signature has been defined (*St Croix et al., 2000*).

Until recently, the study of intra-organ EC heterogeneity has generally been limited to a comparison of large artery, large vein, and microvascular ECs, as these distinctions correspond to the vessel classes that can be physically separated (*Chi et al., 2003*). This technical limitation is circumvented by scRNA-seq, which provides a broad sampling of the EC population that is independent of vessel size and structure, enabling an unbiased clustering of cell classes based only on gene expression profiles. A recent scRNA-seq study of adult mouse brain vasculature provided the first molecular profile of vascular zonation along the arteriovenous axis (*Vanlandewijck et al., 2018*). The data and analyses presented here both corroborate the concept of an EC continuum and extend the findings of Vanlandewijck and colleagues to the early postnatal brain vasculature. In particular, the present work provides strong support for a model in which proliferative ECs are more closely related to venous ECs and tip cells are more closely related to arterial ECs. The data also reveal an unexpected heterogeneity among capillary ECs, with distinct subpopulations that reflect a more venous-like or more arterial-like transcriptional signature. The molecular mechanisms that orchestrate the development of these EC properties remain largely unknown.

## A genomic view of canonical Wnt signaling and CNS vascular development

The earliest evidence for parenchymal signaling to CNS ECs came from embryonic transplantation experiments in which coelomic cavity ECs acquired BBB characteristics when they invaded pieces of transplanted brain, and brain ECs lost BBB characteristics when they invaded pieces of transplanted mesoderm (*Stewart and Wiley, 1981*; *Risau et al., 1986*). Subsequent co-culture experiments showed that astrocytes, which send end-foot processes to contact ECs and pericytes, can induce BBB properties in non-CNS ECs and can maintain BBB properties in long-term cultures of CNS-derived ECs (*Hayashi et al., 1997*); reviewed in *Helms et al., 2016*.

At present, canonical Wnt signaling is the only well-characterized cell-cell signaling pathway that promotes BBB formation and maintenance. In the developing and adult CNS, canonical Wnt ligands are secreted by glia (and possibly neurons) and activate receptors on ECs. In mice, EC-specific loss of beta-catenin leads to abortive CNS angiogenesis and loss of BBB integrity and BBB markers; similarly, double knockout of *Wnt7a* and *Wnt7b* in the embryonic CNS inhibits CNS angiogenesis and suppresses the expression of BBB markers (*Liebner et al., 2008*; *Stenman et al., 2008*; *Daneman et al., 2009*; *Zhou et al., 2014*). Nearly identical phenotypes are observed with mutations in the genes coding for GPR124 and RECK, essential receptor cofactors for WNT7A/WNT7B signaling (*Kuhnert et al., 2010*; *Anderson et al., 2011*; *Cullen et al., 2011*; *Zhou and Nathans, 2014*; *Posokhova et al., 2015*; *Cho et al., 2017*). In the retina, analogous phenotypes are produced by mutations in the genes coding for NORRIN, FRIZZLED4, and TSPAN12 - the ligand, receptor, and

co-activator components of a canonical Wnt signaling system that operates in parallel with the WNT7A/WNT7B system (*Xu et al., 2004*; *Junge et al., 2009*; *Ye et al., 2009*; *Wang et al., 2012*).

The present work extends our understanding of canonical Wnt signaling and BBB development in several ways. First, the location of LEF/TCF motifs in candidate CNS EC-specific CREs permits an initial assessment of which CNS EC genes might be direct targets of canonical Wnt-regulation. Second, the expression of the canonical Wnt reporter exclusively in ECs within and on the surface of the embryonic CNS argues that canonical Wnt signaling does not function as a widely distributed and permissive signal for CNS EC development but may, by virtue of its spatial localization, determine the spatial distribution of the CNS EC fate. The simplest explanation for the observed spatial distribution in canonical Wnt signaling is that it corresponds to the distribution of the relevant ligands: WNT7A, WNT7B, and NORRIN. Third, the nearly identical CNS-specific distribution of ZIC3-expressing ECs - together with (1) the presence of LEF/TCF motifs in candidate CREs near the Zic3 gene, (2) the enrichment of ZIC3 motifs in candidate CNS EC-specific CREs, and (3) the dependence of ZIC3 accumulation on NORRIN/FRIZZLED4 signaling in the developing retinal vasculature (*Wang et al., 2012*) - strongly suggests that ZIC3 is a downstream effector of the canonical Wnt-controlled EC gene expression program.

In summary, we have dissected from a set of candidate CREs common to many cell types, a subset of candidate CREs that are specific to vascular ECs, and an even smaller subset of candidate CREs that are distinctive for tissue-specific EC subtypes and that are likely to control tissue-specific EC gene expression. The genomic resources and analytical approaches developed here should help in identifying the full network of TFs and TF-binding sites that comprise the downstream effectors of parenchyma-derived signals for EC differentiation and specialization.

# Materials and methods

**Key resources table**

| Reagent type (species) or resource | Designation | Source or reference | Identifiers | Additional information |
|---|---|---|---|---|
| Genetic reagent (*Mus musculus*; Male) | Tie2-GFP | The Jackson Laboratory | Stock No: 003658; RRID:IMSR_JAX:003658 | |
| Genetic reagent (*M. musculus*; Male) | Tie2-Cre | The Jackson Laboratory | Stock No: 008863 | |
| Genetic reagent (*M. musculus*; Male) | R26-Tcf/Lef-LSL-H2B-GFP-6xMYC | PMID: 28803732 | | |
| Antibody | anti-CD11b BV421 (rat monoclonal) | Biolegends | Cat No: 101235; RRID:AB_10897942 | 1:1000 |
| Antibody | anti-ICAM2 (rat monoclonal) | BD Biosciences | Cat No: 553326 | 1:300 |
| Antibody | anti-6xMYC (chicken polyclonal) | PMID: 24411735 | | 1:10000 |
| Antibody | anti-GLUT1 (rabbit polyclonal) | Thermo Fisher Scientific | RB-9052 | 1:500 |
| Antibody | anti-ERG (rabbit polyclonal) | Cell Signaling | A7L1G | 1:500 |
| Antibody | anti-dopa decarboxylase (goat polyclonal) | R and D Systems | AF3564 | 1:500 |
| Antibody | anti-ZIC3 (rabbit polyclonal) | PMID: 23217714 | | 1:50 |
| Other | Alexa Fluor 594-conjugated GS Lectin | Thermo Fisher Scientific | L21416 | 1:200 |
| Peptide, recombinant protein | Tn5 transposase | Illumina | FC-121–1030 | |

*Continued on next page*

*Continued*

| Reagent type (species) or resource | Designation | Source or reference | Identifiers | Additional information |
|---|---|---|---|---|
| Commercial assay or kit | Worthington Papain Dissociation Kit | Worthington Biochemical Corporation | LK003160 | |
| Commercial assay or kit | Rneasy Micro Plus Kit | Qiagen | 74034 | |
| Commercial assay or kit | Dneasy Blood and Tissue Kit | Qiagen | 69504 | |
| Commercial assay or kit | MinElute GelExtraction Kit | Qiagen | 28604 | |
| Commercial assay or kit | Agencourt AMPure XP beads | Beckman Coulter | A63880 | |
| Commercial assay or kit | EZ DNA Methylation-Direct Kit | Zymo | D5021 | |
| Software, algorithm | RSEM | PMID: 21816040 | RRID:SCR_013027 | |
| Software, algorithm | DESeq2 | PMID: 25516281 | RRID:SCR_015687 | |
| Software, algorithm | Bowtie2 | PMID: 22388286 | | |
| Software, algorithm | MACS2 | PMID: 18798982 | | |
| Software, algorithm | DiffBind | PMID: 22217937 | RRID:SCR_012918 | |
| Software, algorithm | deepTools | PMID: 27079975 | | |
| Software, algorithm | HOMER | PMID: 20513432 | | |
| Software, algorithm | Monocle | PMID: 24658644 | | |
| Software, algorithm | Methylpy | PMID: 26030523 | | |
| Software, algorithm | BEDTools | PMID: 20110278 | RRID:SCR_006646 | |

## Mice

P7 homozygous *Tie2-GFP* mice (*Motoike et al., 2000*); JAX 003658) were used for isolation of tissue-specific ECs. To visualize Wnt activity in ECs, homozygous *Tie2-Cre* mice (*Kisanuki et al., 2001*); JAX 008863) were crossed to homozygous *R26-Tcf/Lef-LSL-H2B-GFP-6xMYC* mice (*Cho et al., 2017*). To control for the possibility of sex-dependent differences, male mice were used for RNA-seq, ATAC-seq, and MethylC-seq. All mice were housed and handled according to the approved Institutional Animal Care and Use Committee (IACUC) protocol MO16M367 of the Johns Hopkins Medical Institutions.

## EC isolation

ECs were isolated as previously described with some modifications (*Daneman et al., 2010*; *Zhang et al., 2014*). Reagents used included the Worthington Papain Dissociation System (LK003160, Worthington Biochemical Corporation, Lakewood, NJ), Dulbecco's PBS (DPBS, 14287072, Thermo Fisher Scientific, Waltham, MA), trehalose (T0167-10G, MilliporeSigma, Burlington, MA), bovine serum albumin (A7906, Sigma-Aldrich, Burlington, MA), 20 um cell strainers (Cell-Trics, Sysmex Partec, Gorlitz, Germany), and anti-CD11b BV421 (101235, Biolegends, San Diego, CA). For liver, lung, and kidney, a single P7 *Tie2-GFP* mouse typically yielded enough tissue for FACS sorting. For P7, two mice were used. Mice were euthanized and tissues of interest were rapidly

placed into Dulbecco's PBS (DPBS), minced using a razor blade, and enzymatically dissociated in 2.5 ml of papain solution (20 units papain per ml; 1 mM L-cysteine; 0.5 mM EDTA; 5% trehalose; 100 units of DNase per ml) for 1 hr at 37° Celsius. Following this incubation, ovomucoid inhibitor/BSA solution reconstituted at 10 mg per ml was added at 1:10 dilution. The dissociated tissue was gently triturated to form a single-cell suspension. This suspension was then layered on top of a solution consisting of 5 mg per ml ovomucoid inhibitor/BSA and 5% trehalose and centrifuged at 70 x g for 5 min at room temperature. The cell pellet was resuspended in 0.02% BSA and 5% trehalose solution (FACS buffer) and subsequently filtered through a 20 um mesh filter. After another round of centrifugation at 300 x g for 5 min, the cells were resuspended in FACS buffer. Cells were incubated with anti-CD11b BV421 (a macrophage marker) at 1:1000 dilution for 30 min at room temperature. Following antibody staining, cells were pelleted at 300 x g and washed twice with 1 ml of FACS buffer.

Cells were sorted using a MoFlo XDP Sorter (Beckman Coulter, Brea, CA). Viable ECs were defined as GFP positive, propidium iodide negative, and CD11b BV421 negative. For RNA-seq, viable ECs (GFP-positive) and viable parenchymal cells (GFP-negative) were sorted directly into QIAGEN Buffer RLT Plus. For single-cell RNA-seq, ATAC-seq, and MethylC-seq, viable ECs were sorted into FACS buffer for downstream sample preparation.

## Immunohistochemistry

E13.5 embryos were immersion fixed in 4% paraformaldehyde (PFA) overnight at 4° Celsius. For P7 brain, liver, lung, and kidneys, mice were deeply anesthetized and then the tissues were fixed by intracardiac perfusion with PBS followed by 4% PFA in PBS. Tissues were further fixed by immersion in 4% PFA in PBS overnight at 4°C. Embryos and tissues were embedded in 3% low-melting point agarose and vibratome sectioned at a thickness of 120 um. The following reagents were used: DAPI, rat anti-ICAM2 (1:300; 553326, BD BioSciences, San Jose, CA), chicken anti-6xMYC (1:10,000; *Wu et al., 2014*), rabbit anti-GLUT1 (1:500; RB-9052, Thermo Fisher Scientific), rabbit anti-ERG (1:500; A7L1G, Cell Signaling Technology, Danvers, MA), goat anti-dopa decarboxylase (1:500; AF3564, R and D Systems, Minneapolis, MN), Alexa Fluor 594-conjugated GS Lectin (1:200; L21416; Thermo Fisher Scientific), rabbit anti-ZIC3 (1:50; *Wang et al., 2012*). For the images in *Figure 6A and C*, the unit for quantification was a rectangle 410 um high x 355 um wide, and quantifications were performed using three adjacent rectangles overlying the CNS and three adjacent rectangles overlying the mesoderm (i.e. non-CNS) region. Within each rectangle, the number of MYC+, ERG+, or ZIC3 +nuclei were counted and the vasculature was traced in Adobe Illustrator and the lengths of the traced lines were quantified by using ImageJ to count the resulting pixels.

## Whole mount retina in situ hybridization

Flat-mount retina ISH was performed as previously described (*Rattner et al., 2013*). Briefly, enucleated eyes were fixed in 2% PFA for 5 min at room temperature and then transferred to hypertonic PBS (2x PBS) for 5 min at room temperature. Retinas were dissected and four radial incisions were made. With the vitreal surface facing up, buffer was slowly removed until retinas were flat. In order to fix the flattened retinas, methanol cooled to −20°C was added dropwise. For ISH, fixed retinas were first transferred to 4% PFA for 5 min at room temperature, then washed three times with RNAase-free PBS with 0.1% Tween-20 (PBST), digested with 20 ug/mL Proteinase K (P2308, Sigma-Aldrich) for 15 min at room temperature, fixed in 0.2% glutaraldehyde with 4% PFA, washed three times with PBST, and finally transferred to hybridization solution (50% formamide, 5x saline-sodium citrate buffer, 5 mM EDTA, 2% Blocking Reagent (11096176001, Sigma-Aldrich), 50 ug/mL yeast RNA, 100 ug/mL heparin, 0.5% CHAPS, and 0.1% Triton X-100) at 65°C. After 2 hr of prehybridization, digoxigenin-labeled riboprobes were added, and hybridization proceeded overnight at 65°C. After a series of washes (*Rattner et al., 2013*), retinas were blocked for 2 hr at room temperature in maleic acid buffer with 0.1% Tween-20 (MABT), 2% Blocking Reagent, and 10% normal goat serum. Then, retinas were incubated with anti-digoxigenin-AP Fab fragments (1:1000; 11093274910, Sigma-Aldrich) overnight at 4°C. The following day, retinas were washed four times for 1 hr each at room temperature and then overnight at 4°C in MABT. Retinas were washed three times in AP buffer with 0.1% Tween-20 for 5 min at room temperature, and then they were stained for 1–4 hr with AP buffer and nitro-blue tetrozolium (NBT; 350 ug/mL) and 5-bromo-4-chloro-3-indolyl phosphate (BCIP; 175 ug/mL). After staining, retinas were washed twice with PBST, post-fixed for 5 min with 4% PFA,

washed twice with PBST, and stained with Alexa Fluor 594-conjugated GS Lectin (1:100) overnight at 4°C. For imaging, retinas were dehydrated by transferring the retinas between increasing concentrations of ethanol (25% in PBST, 50% in PBST, 75% in PBST, and 100%). Finally, retinas were cleared in benzyl benzoate:benzyl alcohol (2:1). For presentation, the purple ISH signal was converted to green, and the GS Lectin signal was converted to magenta.

## Microscopy

Confocal images were captured with an LSM700 (Zeiss, Jena, Germany) microscope. ISH images were captured with a Zeiss Imager Z1 Microscope using Zeiss AxioVision 4.6 software. Image processing was performed using FIJI (*Schindelin et al., 2012*).

## RNA and DNA sample preparation

RNA was extracted from an aliquot of dissociated tissue prior to filtering and antibody staining and from GFP-positive and GFP-negative FACS sorted cells using the RNeasy Micro Plus kit (74034, QIAGEN, Venlo, Netherlands). DNA for MethylC-seq was prepared from GFP-positive FACS-sorted cells using the DNeasy Blood and Tissue kit (69504, QIAGEN). For ATAC-seq,~50,000 GFP-positive FACS-sorted cells were resuspended in ice-cold lysis buffer (0.25 M sucrose, 25 mM KCl, 5 mM $MgCl_2$, 20 mM Tricine-KOH, 0.1% Igepal CA-630) and immediately centrifuged at 500 x g for 10 min at 4°C to prepare nuclei. The resulting nuclear pellet was resuspended in a 50 ul reaction volume in Tn5 transposase and transposase reaction buffer (FC-121–1030, Illumina Inc., San Diego, CA) and the tagmentation reaction was incubated at 37°C for 30 min.

## Library preparation and sequencing

Each RNA-seq, ATAC-seq, and MethylC-seq analysis was conducted on two biological replicates except for brain EC ATAC-seq, which was conducted on three biological replicates. Single-cell RNA-seq was conducted with a single sample. Libraries for RNA-seq and ATAC-seq were prepared as previously described, with minor modifications (*Buenrostro et al., 2015*; *Lister et al., 2013*; *Mo et al., 2015*; *Mo et al., 2016*). For RNA-seq, total RNA was converted to cDNA and amplified (Ovation Ultralow System V2-32, 0342HV, NuGEN Technologies, San Carlos, CA). Amplified cDNA was fragmented, end-repaired, linker-adapted, and single-end sequenced for 75 cycles on a Next-Seq500 (Illumina Inc.). Tagmented DNA was purified using QIAGEN MinElute GelExtraction kit (28604, Qiagen). ATAC-seq libraries were PCR amplified for 11 cycles. Agencourt AMPure XP beads (A63880, Beckman Coulter) were used to purify ATAC-seq libraries, which were then paired-end sequenced for 36 cycles on a NextSeq500. MethylC-seq library preparation discussed below.

## Data analysis

Most data analysis was performed as previously described (*Mo et al., 2016*). For basic data processing, exploration, and visualization, we used deepTools (*Ramírez et al., 2016*), BEDTools (*Quinlan and Hall, 2010*), RStudio (*Team RS, 2016*), the tidyverse collection of R packages (*Wickham, 2017*), ggplot2 (*Wickham, 2009*), pheatmap (*Kolde, 2015*), and custom scripts. Reads were aligned to the mm10 genome using Bowtie2 (*Langmead and Salzberg, 2012*). The Integrative Genomics Viewer (IGV) was used to visualize bigwig files generated by deepTools bamcoverage (*Robinson et al., 2011*; *Thorvaldsdóttir et al., 2013*).

## RNA-seq

Three different sets of RNA-seq data were generated from papain dissociated P7 *Tie2-GFP* brain, liver, lung, and kidney: (1) total tissue; (2) GFP-negative FACS-sorted cells; and (3) GFP-positive FACS-sorted cells. RSEM version 1.3.0 (*Li and Dewey, 2011*) was used to align reads with Bowtie2 to the mm10/GRCm38 genome and calculate transcript expression using Ensembl annotation (*rsem-calculate-expression* –bowtie2 –estimate-rspd –append-names –output-genome-bam –sort-bam-by-coordinate). Differentially expressed genes were identified using EBSeq version 1.18.0 (*Leng and Kendziorski, 2015*). To filter out background transcripts from surrounding parenchymal cells, we determined a set of EC-enriched transcripts for each tissue by comparing GFP-positive and GFP-negative sorted cells. A transcript was considered EC-enriched if it met the following three criteria: (1) a minimum two-fold enrichment in GFP-positive compared to GFP-negative samples; (2) a

posterior probability of differential expression (PPDE) greater than or equal to 0.95 [PPDE = (1 - false discovery rate)]; (3) relative expression greater than or equal to 10 transcripts per million (TPM) in both biological replicates. A gene was considered to be differentially expressed between EC subtypes if it met the following criteria: (1) EC-enriched with minimum of two-fold enrichment between one subtype and all three other subtypes; (2) a PPDE greater than or equal to 0.95; (3) a TPM value greater than or equal to 10 in both biological replicates. Principal component analysis was performed on 'regularized log'-transformed data using the DESeq2 *rlog* and *plotPCA* function (*Love et al., 2014*). Gene ontology enrichment analysis was performed on EC tissue-specific genes using the online enrichment analysis tool (*Mi et al., 2017*) from the Gene Ontology Consortium.

## ATAC-seq

We aligned ATAC-seq data using Bowtie2 (Version 2.3.2 t -X 2000 –no-mixed –no-discordant) and then removed duplicate reads (*picard MarkDuplicates*). Peaks were called using MACS2 (Version 2.1.1.20160309 *callpeak* –nomodel –keep-dup all –shift −100 –extsize 200 –call-summits) (*Zhang et al., 2008*). Peaks were then filtered for fold-change >2 and -log(qvalue)>2. ATAC-seq peaks visualized on the browser were called using all fragment sizes. To determine the fraction of mapped reads in called peaks, we used deepTools *multiBamSummary* (BED-file –outRawCounts – samFlagInclude 64). Samtools *view* (-s) was used to subsample reads. For all downstream analyses, ATAC-seq peaks were called using paired-end sequencing fragments < 100 bp in length, which are enriched for nucleosome-free regions.

To identify differential ATAC-seq peaks between ECs isolated from brain, liver, lung, and kidney, we used DiffBind (*Stark and Brown, 2011*; *Ross-Innes et al., 2012*) with both DESEQ2 and EdgeR (*Robinson et al., 2010*; *McCarthy et al., 2012*) methods. DiffBind was used to develop a set of consensus peaks between replicates using the requirement that peaks must be in at least two thirds of the replicates (minOverlap = 0.66). From this new set of peaks, a consensus peakset was generated corresponding to the discrete set of peaks identified between all samples being compared (minOverlap = 1). Log$_2$ transformed normalized read counts in each sample are then tabulated over each peak region and plotted. To retrieve a set of high-confidence, cell type-enriched peaks, differential peak analysis was carried out using DESEQ2 and EdgeR. From both sets of analysis, we filtered for peaks with an absolute fold difference >2 and FDR < 0.05. In R, the union of these two sets of peaks was used to generate a final set of differentially enriched peaks for each sample. Principal component analysis was performed on 'regularized log'-transformed data using the DESeq2 *rlog* and *plotPCA* function.

## Methylome sequencing

DNA methylome libraries were prepared using a modified snmC-seq protocol adapted for bulk DNA samples (*Luo et al., 2017*). 20 ng of purified genomic DNA with 0.5% unmethylated lambda DNA spike-in (D1521, Promega) was bisulfite converted using the EZ DNA Methylation-Direct Kit (D5021, Zymo) following the product manual and was eluted in 10 μl M-Elution buffer. 9 μl converted DNA was mixed with 1 μl P5L_random oligo (5 μM,/5SpC3/TTCCCTACACGACGCTC TTCCGATCTNNNNNNNNNN, IDT) followed by heat denaturing at 95℃ for 3 min using a thermocycler. The samples were chilled on ice for 2 min and mixed with 10 μl enzyme mix containing 2 μl of Blue Buffer (B0110, Enzymatics), 1 μl of 10 mM dNTP (N0447L, NEB), 1 μl of Klenow exo- (50 U/μl, P7010-HC-L, Enzymatics) and 6 μl H2O. The samples were incubated using a thermocycler with the following program: 4℃ for 5 min, ramp up to 25℃ at 0.1℃/s, 25℃ for 5 min, ramp up to 37℃ at 0.1℃/s, 37℃ for 60 min, 4℃. A 3 μl enzyme mix containing 2 μl of Exonuclease 1 (20U/μl, X8010L, Enzymatics) and 1 μl of Shrimp Alkaline Phosphatase (rSAP, M0371L, NEB) was added to the samples. The samples were incubated at 37℃ for 30 min using a thermocycler. Samples were cleaned-up using 0.8x SPRI beads and eluted with 10 μl M-Elution buffer. Eluted samples were heated at 95℃ for 3 min using a thermocycler and were chilled on ice for 2 min. Samples were mixed with 10.5 μl Adaptase master mix containing 2 μl Buffer G1, 2 μl Regent G2, 1.25 μl Reagent G3, 0.5 μl Enzyme G4, 0.5 μl Enzyme G5 and 4.25 μl M-Elution buffer (Accel-NGS Adaptase Module for Single Cell Methyl-Seq Library Preparation, 33096, Swift Biosciences). Samples were incubated at 37℃ for 30 min, 95℃ for 2 min and 4℃ using a thermocycler. 1 μl 30 μM P5 indexing primer and 5 μl 10 μM P7 indexing primer and 25 μl KAPA HiFi HotStart ReadyMix, (KK2602, KAPA BIOSYSTEMS) were

added into each sample followed by thermocycling using the following program - 95℃ for 2 min, 98℃ for 30 s, 10 cycles of (98℃ for 15 s, 64℃ for 30 s, 72℃ for 2 min), 72℃ for 5 min and then 4℃. PCR products were cleaned up using 0.8x SPRI beads twice. Methylome libraries were sequenced using an Illumina HiSeq 4000 instrument with 5% PhiX DNA spike-in.

Adaptor sequences were trimmed from sequencing reads using Cutadapt 1.11 with parameters -f fastq -q 20 m 50 -a AGATCGGAAGAGCACACGTCTGAAC -A AGATCGGAAGAGCGTCGTG TAGGGA. We further trimmed 10 bp from both 5'- and 3'- ends of both R1 and R2 reads with parameters -f fastq -u 10 u −10 m 30. Trimmed R1 and R2 reads were mapped to the mm10 reference genome as single-end reads using bismark 0.14.4 with parameter –bowtie2. –pbat option was used for mapping R1 reads. Mapped reads were sorted using Samtools 1.3 followed by removing duplicate reads using Picard 1.141 MarkDuplicates with parameter REMOVE_DUPLICATES = true. Reads were further filtered by MAPQ > 20 using samtools view with parameter -bhq20. Methylation level at each cytosine was summarized using call_methylated_sites from Methylpy package (https://github.com/yupenghe/methylpy/).

### Methylome analysis and identification of ECTS-hypo-DMRs, UMRs, LMRs, and large hypomethylation features

DMRs were identified as previously described in (*Mo et al., 2015*) using *DMRfind* from the Methylpy package with parameters num_sims = 3000, num_sig_tests = 100, use_mc_status = False, dmr_max_dist = 250, sig_cutoff = 0.01. DMRs showing hypo-methylation in both biological replicates were used for further analyses.

UMRs and LMRs were identified using MethylSeekR (*Burger et al., 2013*) with m (methylation level)=0.3% and 5% FDR. DMVs were identified as UMRs > 3 kb. To identify large ECTS-hypo-DMRs, all ECTS hypo-DMRs with inter-DMR distances < 1 kb were merged (bedtools merge -d 1000). Large ECTS-hypo-DMRs were defined as merged ECTS-hypo-DMRs>2 kb.

For MethylC-seq scatterplots in *Figure 2A* and *Figure 2—figure supplement 1*, *multiBigwigSummary* (BED-file –outRawCounts) of deepTools was used to compute the mean fraction of mCG in a 5 kb window 3' of the TSS of protein-coding genes. Gene ontology analysis of shared DMVs was performed using Genomic Regions Enrichment of Annotations Tool (GREAT) (*McLean et al., 2010*). To compare the degree of methylation between EC subtypes at DMVs overlapping differentially expressed TF genes, we applied the following steps: (1) within each EC subtype, all DMVs in a 20 kb window 5' to the TSS of the gene of interest were joined into a single contiguous block, (2) overlapping DMVs between EC subtypes were merged to generate a genomic interval that spanned the greatest extent of all the DMVs, (3) within the merged interval, the fraction mCG methylation was calculated for each EC subtype. Computation was performed using *multiBigwigSummary*. In *Figure 4—figure supplement 3B and C*, *computeMatrix* (reference-point –referencePoint center) was used to compute either the degree of methylation at ATAC-seq peaks or the ATAC-seq signal at UMRs, LMRs, and DMRs as a function of distance from the center of regions. Then the deepTools function *plotProfile* was used to generate output data in table form for plotting in R. The ATAC-seq signal at hypomethylated features was upper quartile normalized using the *uqua* function of the R package NOISeq (*Tarazona et al., 2011*; *Tarazona et al., 2015*). For *Figure 4—figure supplement 3C*, each line was translated vertically so that the value at position −1500 bp is 0.

### Feature overlap analysis

*Bedtools intersect* (-u) was used to determine features that overlapped with other features by greater than or equal to 1 bp. *Bedtools window* (-w 100000) was used to identify CREs within 100 kb of ECTSGs. For *Figure 2B*, statistically significant enrichment was tested by hypergeometric distribution using MATLAB *hygecdf* function as previously described in *Mo et al., 2015*. For *Figure 4—figure supplement 2C*, statistically significant enrichment was tested with *hygecdf(number of category i feature overlapping with category j DEG, sample size of either all hypo-DMRs and ECTS-hypo-DMRs or all differential APs and ECTSAPs, number of features (either DMRs or APs) that overlap with category j DEG, sample size of category i feature, 'upper')*.

## Transcription factor DNA binding motif enrichment analysis

TF binding motifs in candidate regulatory regions (ECTSAPs and ECTS-hypo-DMRs) were identified and analyzed using the HOMER suite of tools for motif discovery (*Heinz et al., 2010*), in particular, *findMotifsGenome.pl* (-size given). HOMER searches for enriched motifs using a de novo strategy of looking for enriched k-mers in target regions compared to a set of random background genomic regions. HOMER also maintains a custom database of known motifs based on published ChIP-Seq data sets. From the set of known motifs, we filtered for motifs exhibiting a q-value <0.001 and a fold enrichment (defined as the % of target regions with the motif divided by the % of background regions with the motif) >2 for each sample. These were considered to be statistically significant. De novo-enriched motifs could typically be matched to a motif from the list of enriched known motifs. Because many members of a TF family share a similar core motif, we chose one representative member of each TF family motif to display in motif heatmaps (*Figures 5B and F*). *AnnotatePeaks.pl* (-m -size 1000 -hist 5) was used to generate histograms of enriched motifs, and *annotatePeaks.pl* (-center -size given -multi) was used to center candidate regions on instances of motifs. To identify the number of features containing a given motif, we used *annotatePeaks.pl* (-m -nomotifs -noann -nogene). To determine statistically significant enrichment of a motif (as in *Figure 5—figure supplement 1B,C and E*), we used *hygecdf(number of category i feature overlapping with category j motif, sample size of either ECTS-hypo-DMRs or ECTSAPs within 100 kb of ECTSGs, number of features (either DMRs or APs) that contain category j motif, sample size of category i feature, 'upper')*.

## Droplet-based single-cell RNA-seq of brain ECs

FACS purified GFP-postive P7 brain ECs from a *Tie2-GFP* mouse were processed through the Gem-Code Single Cell Platform using the GemCode Gel Bead, Chip, and Library Kits (10X Genomics, Pleasanton, CA) as per the manufacturer's protocol. In brief, papain-dissociated ECs were sorted into 0.4% BSA–DPBS, and approximately 10,000 FACS-purified singlet brain ECs were added to the channel with a final recovery rate of 4,257 cells. ECs were then partitioned into gel beads in emulsion in the GemCode instrument for cell lysis and barcoded oligo-DT priming and reverse transcription of poly-adenylated RNA. This was followed by PCR amplification, shearing, and 5′ adaptor and sample index attachment. Libraries were sequenced on an Illumina HiSeq 2500.

## Analyses of single EC transcriptomes

scRNA-seq data were pre-processed using the cellranger pipeline (v2.0; 10x Genomics) with default settings. The normalized geneXcell matrix was used as input for the Monocle R/Bioconductor package (*Trapnell et al., 2014*). Log-normalized expression estimates (with a pseudocount of 1) were used as input for a modified dpFeature analysis (Monocle). Briefly, high-dispersion genes were identified as those genes with a residual greater than 1 * the estimated mean-variance split using the 'estimateDispersions' method in Monocle. Non-endothelial-specific transcripts in the high-dispersion gene set, defined as transcripts enriched >2 fold by bulk RNA-seq in the FACS-purified GFP-negative fraction compared to the GFP-positive fraction, likely derived from high-abundance transcripts released and solubilized by lysis of non-ECs during the preparation of the single-cell suspension. 311 non-EC specific genes were excluded from the high-dispersion gene set and removed from subsequent analyses. For preliminary cell clustering, principal component analysis (PCA) was performed on the high-variance genes and components 2–10 were visualized in two dimensions using t-Distributed Stochastic Neighbor Embedding (t-SNE) (*Van Der Maaten and Hinton, 2008*; *Krijthe, 2015*; *Macosko et al., 2015*). Component one was excluded as it was determined to primarily reflect capture efficiency (num_genes_expressed). Initial clusters were identified from the 2D embedding using Mclust. To refine the list of genes contributing to the initial clustering, we performed a Monocle differential gene test with respect to cluster assignment, accounting for total mRNAs in both full and reduced models as well. The top 1200 genes with significant differential expression across clusters were then selected and used as input for a second PCA and components 2–8 were reduced into two dimensions using tSNE. Final cluster assignments were determined using Mclust on the updated 2D embedding. Clusters were annotated based on the expression of canonical EC subtype markers and a differential gene test between the subtypes was performed using the R package Monocle. The final six EC clusters conformed well to the expression patterns of known markers for arterial, venous,

tip, capillary and mitotic cells. Cell trajectory analyses were carried out in Monocle on the same 1200 differentially expressed genes used for the second PCA.

## Acknowledgements

We thank Hao Zhang of the Johns Hopkins Bloomberg School of Public Health Flow Cytometry Core Facility, and Haiping Hao, Linda Orzolek, and Jasmeet Sethi of the Johns Hopkins Medical Institutions Deep Sequencing and Microarray Core Facility for technical assistance.

## Additional information

### Competing interests

Jeremy Nathans: Reviewing editor, *eLife*. The other authors declare that no competing interests exist.

### Funding

| Funder | Grant reference number | Author |
| --- | --- | --- |
| Howard Hughes Medical Institute | | Joseph R Ecker<br>Jeremy Nathans |
| National Eye Institute | R01EY018637 | Jacob S Heng<br>Amir Rattner<br>Jeremy Nathans |
| Arnold and Mabel Beckman Foundation | | Jacob S Heng<br>Jeremy Nathans |
| Eunice Kennedy Shriver National Institute of Child Health and Human Development | F30HD088023 | Mark F Sabbagh |
| National Science Foundation | IOS-1665692 | Loyal A Goff |
| Maryland Stem Cell Research Fund | 2016-MSCRFI-2805 | Loyal A Goff |
| Johns Hopkins University | Science of Learning and Synergy Awards | Loyal A Goff |

The funders had no role in study design, data collection and interpretation, or the decision to submit the work for publication.

### Author contributions

Mark F Sabbagh, Conceptualization, Data curation, Software, Formal analysis, Funding acquisition, Validation, Investigation, Visualization, Methodology, Writing—original draft, Writing—review and editing; Jacob S Heng, Conceptualization, Data curation, Software, Formal analysis, Funding acquisition, Investigation, Visualization, Methodology, Writing—original draft, Writing—review and editing; Chongyuan Luo, Data curation, Software, Formal analysis, Validation, Investigation, Visualization, Methodology, Writing—review and editing; Rosa G Castanon, Joseph R Nery, Validation, Investigation; Amir Rattner, Conceptualization, Resources, Software, Supervision, Methodology, Project administration, Writing—review and editing; Loyal A Goff, Resources, Software, Supervision, Funding acquisition, Visualization, Writing—review and editing; Joseph R Ecker, Resources, Supervision, Funding acquisition, Writing—review and editing; Jeremy Nathans, Conceptualization, Resources, Supervision, Funding acquisition, Methodology, Writing—original draft, Project administration, Writing—review and editing

### Author ORCIDs

Mark F Sabbagh https://orcid.org/0000-0003-1996-5251
Jacob S Heng https://orcid.org/0000-0001-6291-6688
Chongyuan Luo https://orcid.org/0000-0002-8541-0695

Loyal A Goff [iD] http://orcid.org/0000-0003-2875-451X
Joseph R Ecker [iD] http://orcid.org/0000-0001-5799-5895
Jeremy Nathans [iD] http://orcid.org/0000-0001-8106-5460

## Ethics

Animal experimentation: This study was performed in strict accordance with the recommendations in the Guide for the Care and Use of Laboratory Animals of the National Institutes of Health. All mice were housed and handled according to the approved Institutional Animal Care and Use Committee (IACUC) protocol MO16M367 of the Johns Hopkins Medical Institutions.

## Decision letter and Author response

Decision letter https://doi.org/10.7554/eLife.36187.041
Author response https://doi.org/10.7554/eLife.36187.042

# Additional files

## Supplementary files

• Supplementary file 1. Characteristics of each sequencing sample. (A) RNA-seq. (B) ATAC-seq. (C) MethylC-seq
DOI: https://doi.org/10.7554/eLife.36187.029

• Supplementary file 2. Gene expression data. (A-C) Transcript abundances in raw expected read counts (A), cross-sample normalized read counts (B), and TPMs (C). (D) List of EC-enriched transcripts. (E) List of all differentially expressed genes between one or more pairs of EC subtypes. (F) List of ECTSGs
DOI: https://doi.org/10.7554/eLife.36187.030

• Supplementary file 3. Gene ontology (GO) enrichment analysis (A) GO enrichment analysis of brain ECTSGs. (B) GO enrichment analysis of liver ECTSGs. (C) GO enrichment analysis of lung ECTSGs. (D) GO enrichment analysis of kidney ECTSGs.
DOI: https://doi.org/10.7554/eLife.36187.031

• Supplementary file 4. Hypomethylated features in each EC subtype (A-D) UMRs for each EC subtype. (E-H) LMRs for each EC subtype. (I-L) DMRs for each EC subtype. (I'-L') ECTS-hypo-DMRs for each EC subtype. (M-P) DMVs for each EC subtype. (Q-T) ECTS-large hypo-DMRs for each EC subtype. (U) LMRs found only in ECs relative to photoreceptors and cortical neurons. (V) LMRs from (U) that are shared between EC subtypes
DOI: https://doi.org/10.7554/eLife.36187.032

• Supplementary file 5. Numbers underlying heatmaps shown throughout figures.
DOI: https://doi.org/10.7554/eLife.36187.033

• Supplementary file 6. Accessible chromatin peaks in each EC subtype. (A-D) ATAC-seq peaks for each EC subtype called using the full range of ATAC-seq fragment lengths. (E-H) ATAC-seq peaks for each EC subtype called using <100 nt ATAC-seq fragments. (I-L) Differential ATAC-seq peaks (<100 nt) for each EC subtype. (I'-L') ECTSAPs (<100 nt) for each EC subtype. (M) ATAC-seq peaks (<100 nt) found only in ECs relative to photoreceptors and cortical neurons. (N) ATAC-seq peaks from (M) that are shared between EC subtypes.
DOI: https://doi.org/10.7554/eLife.36187.034

• Supplementary file 7. HOMER motif files used in this study. (A) HOMER motif files for enriched k-mers identified in ECTS-hypo-DMRs and ECTSAPs. (B) HOMER motif file used for representative member of TF families. (C) HOMER motif file used for paired ETS:ZIC motif.
DOI: https://doi.org/10.7554/eLife.36187.035

• Supplementary file 8. Top 25 genes for each single-cell RNA-seq cluster. (A) Arterial cluster markers. (B) Capillary-A cluster markers. (C) Capillary-V cluster markers. (D) Mitotic cluster markers. (E) Tip cell cluster markers. (F) Venous cluster markers.
DOI: https://doi.org/10.7554/eLife.36187.036

• Transparent reporting form

DOI: https://doi.org/10.7554/eLife.36187.037

## Data availability

Sequencing data have been deposited in GEO under accession code GSE111839

The following dataset was generated:

| Author(s) | Year | Dataset title | Dataset URL | Database, license, and accessibility information |
|---|---|---|---|---|
| Sabbagh MF, Heng JS, Luo C, Castanon RG, Nery JR, Rattner A, Goff LA, Ecker JR, Nathans J | 2018 | Transcriptional and Epigenomic Landscapes of CNS and non-CNS Vascular Endothelial Cells | https://www.ncbi.nlm.nih.gov/geo/query/acc.cgi?acc=GSE111839 | Publicly available at the NCBI Gene Expression Omnibus (accession no: GSE111839). |

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
