## [Decision Letter]

Thank you for submitting your article "Transcriptional and epigenomic landscapes of CNS and non-CNS vascular endothelial cells" for consideration by *eLife*. Your article has been reviewed by three peer reviewers, including Elisabetta Dejana as the Reviewing Editor and Reviewer #1, and the evaluation has been overseen by Didier Stainier as the Senior Editor.

The reviewers have discussed the reviews with one another and the Reviewing Editor has drafted this decision to help you prepare a revised submission.

Thanks for submitting your manuscript to *eLife*. The two reviewers and I found the paper of interest and certainly important for the advancement of the field.

However, as specified in detail in their comments the reviewers feel that the manuscript requires revision. In particular, the data on in vitro brain endothelial cell culture should be done in a more accurate way. For instance, the role of canonical Wnt in maintenance of the BBB characteristics in vitro should be analyzed in more detail as recommended. Both reviewers 2 and 3 ask for other relevant additions for a more adequate comparison of cultured and freshly isolated endothelial cells. Some relevant references are missing:

Vanlandewicjk et al., 2018; Paolinelli et al., 2013 for in vitro BBB cell culture.

Importantly, Vanlandewicjk et al., should be cited and discussed.

The full reviews are also included for your reference, as they contain detailed and useful suggestions.

Reviewer #1:

This is a particularly complex paper that focus on endothelial heterogeneity in different organs (liver, lung kidney and brain).

The authors made an extensive effort and report interesting data on the transcriptome, accessible chromatin and methylome of the different types of ECs. They identified both shared and organ specific signaling pathways that regulate endothelial specificity.

I believe that the paper contains a lot of important information that was only discussed and hypothesized before but not directly shown such as the correlation between venous markers and cell proliferation or arterial markers and tip cells. Therefore, this paper will certainly add to our knowledge on endothelial cells heterogeneity and plasticity. The text and experiments are particularly detailed and not always easy to digest but it is probably impossible to synthetize so many different and extensive data in a more simple way.

Overall, I am in favor of the publication of this work without any specific change or revision.

Reviewer #2:

This manuscript entitled "Transcriptional and epigenomic landscapes of CNS and non-CNS vascular endothelial cells" analyzes transcriptional and epigenomic differences between four distinct types of ECs (brain, lung, liver and kidney) using RNA-seq, DNA methylation profiles and accessible chromatin regions with ATAC sequencing. In addition, the authors isolate brain ECs from P7 brains and culture them for a week to identify potential changes in RNA-seq, DNA methylome and accessible chromatin between brain ECs in vivo and in vitro that may account for loss of BBB properties in vitro.

Although, multiple studies have now been performed on identification of transcriptional differences between ECs in different organs (brain, lung, liver) and the concept that the vasculature is highly specialized for each organ is well accepted in the field of vascular biology, the study provides some novel and interesting insights into the biology of transcriptional specializations of ECs for distinct organs. This is a tour-de-force study that provides a detailed analysis of differences in DNA methylome and accessible chromatin between distinct cell types.

The study suffers from some drawbacks that need to be addressed in greater detail to enhance the significance of the study.

1) One of the major analyses is the comparison between brain ECs in vivo and in vitro. The authors show nicely that several genes that are activated by canonical Wnt signaling become downregulated within a week in culture. What specific culture conditions were used to culture mouse endothelial cells? Several key studies in the field of vascular biology have shown that addition of serum or ECGS in the media can affect the transcriptional landscape of ECs in culture. Therefore, the findings from the authors about the presence of differences are not surprising. However, can the authors perform a similar analysis using more defined conditions and various EC media to determine which conditions may provide the minimum divergence from the brain ECs? This will be highly valuable for vascular biologists and represent a major advancement in the field.

2) Does addition of Wnts or Wnt agonists to cultured brain ECs restores their properties to a certain extent or is the divergence in culture so extensive that is irreversible? When does this process become irreversible?

3) Several studies have shown that brain or lung ECs in culture retain some tissue-specific identity and expression of certain genes in culture. What happens to some of the BBB-specific genes in cultured brain ECs? The authors do not show in detail BBB specific genes.

4) Do lung ECs diverge to a "generic" EC when cultured for several days? It will be important to test this hypothesis and show the data for another cell type beside the CNS.

5) In Figure 7, the authors show some specific new markers for tip cells or capillary-A, capillary-V markers. A recent study by Betsholtz has shown that there is a zonal expression of several endothelial genes. Therefore, these findings are not surprising. Have the authors validated any of these markers in particular tip cell markers? They only show DCC. It will be important to validate other genes.

The data file provided so far are sufficient.

Reviewer #3:

Sabbagh et al., compare the transcriptional and epigenetic landscapes of endothelial cells in different organs: brain, liver, lung, and kidney. This is the first study to perform both RNA-seq, MethylC-seq and ATAC-seq of ECs in different organs, and the study also includes some brain EC vascular zonation analyses (a novel and timely topic). Lastly, the study compares expression of freshly isolated ECs and in vitro ECs cultured 8 days, but flaws exist with the design of this final experiment (see below comments). While the present study is interesting and has some novelty overall, the study could be improved by addressing the comments below:

1) In vivo and in vitro transcriptional and epigenetic data (Figure 9 and Figure 10) are generated from two different strains of mice at two different developmental ages (in vivo: *Tie2-GFP* mice at P7; in vitro: C57BL/6 mice at P28-56); then the authors conclude from this data that in vitro culture affects gene expression. For this comparison to be valid, the authors should similarly analyze in vivo and in vitro cultured cells from the same genetic mouse strain at the same age, and then determine the degree to which in vitro culture shifts gene expression. The time in culture is also important as it might affect the results. For instance, after how many population doubling times EC were studied? With increased timing in culture ECs might turn towards senescent-like phenotype. It would be really cool to show for how long they can sustain close to their in vivo determine profile. This is also why comparing mice at similar age may be important especially as a recent study in Nature 2018 suggests that some endothelial cells do not fit the A-V zonation pattern and exhibit high expression of ribosomal protein transcripts, which indicates that protein synthesis occurs throughout the A-V axis.

2) The entire study uses 2 biological replicates consisting of 2 total number of mice for liver, kidney and lung and 4 total mice (2 mice pooled per replicate) for brain transcriptional and epigenetic data (except brain ATAC-seq that was conducted with 3 biological replicates). We think that 2 biological replicates may not be sufficient.

3) The authors may consider developing a website to enable readers to more easily access the study's transcriptional and epigenetic data, as many other groups have done (see for example Zhang et al., 2014; He et al., 2016; Vanlandewijck et al., 2018).

4) The manuscript's overall interpretation, readability and data presentation could be improved. For example, the authors present the degree of methylation of differentially expressed EC genes in each organ expressed as gene browser snapshot and bar/scatter plots (Figure 2—figure supplement 2), but no statistical analyses are performed, and the methylation pattern interpretation is not always clear. Moreover, the many gene browser snapshots presented throughout the figures are not reader friendly and do not clearly demonstrate the point the authors are trying to make in the text.

5) In addition to reporting Gene Ontology (GO) pathways shared by brain, liver, lung and kidney (Figure 2—figure supplement 1E), the authors should also include GO pathways terms that are unique to ECs in each organ.

6) The authors present heatmaps depicting gene expression and mean fraction of methylated CG in the gene body for genes within four HOX clusters presented anterior to posterior (Figure 3C). Many of the genes shown lack an association between high gene expression and high methylation or low gene expression and low methylation. What is the interpretation of this figure? This should be added and adequately discussed in the text.

7) It is quite surprising the authors do not cite a recent Nature paper that is the first to present transcriptional analysis of vascular zonation profiles of brain endothelial cells and mural cells (Vanlandewicjk et al., Nature, 2018). This paper should be cited and discussed. In regard to this paper and clustering of different brain endothelial types along the A-V axis – arterial, capillary and venous (also shown in the authors Figure 7 and Figure 8), do the authors know which endothelial cell clusters are represented by their in vitro analysis? This also could be the source for variation between in vitro and in vivo findings, as in vitro ECs might have differentially enriched clusters of cells compared to in vivo, depending on the source of origin.

8) The EC Methyl-seq correlation between different organs is very good: r=0.95-0.97 (Figure 2A; Figure 2—figure supplement 1B) and r=0.90-0.92 (Figure 4—figure supplement 3), compared to the EC RNA-seq (normalized counts) correlation between different organs (r=0.45-0.61) (Figure 1B). Why is this? The authors may want to elaborate on this in the discussion.

9) It would be helpful if the authors can provide quantification for their immunohistochemistry data (e.g., Figure 6).

10) This study only uses male mice. This is a limitation given the NIH's urge to be inclusive of both genders.

[Editors' note: further revisions were requested prior to acceptance, as described below.]

Thank you for resubmitting your work entitled "Transcriptional and epigenomic landscapes of CNS and non-CNS vascular endothelial cells" for further consideration at *eLife*. Your revised article has been favorably evaluated by Didier Stainier (Senior Editor), a Reviewing Editor, and two reviewers.

All of the reviewers agree that the paper has been improved significantly after revision. However, some doubts still remain that need to be addressed before acceptance. These are related to the comparison of brain derived endothelial cells in vivo or in culture conditions in vitro.

Specifically, a more accurate study on the culture conditions used seems important to draw conclusions. It has been known for many years that when brain endothelial cells are isolated and cultured they lose their BBB properties quite rapidly, unless they are maintained in co-culture with astrocytes (see, for instance, papers from Cecchelli et al., and many others) or when Wnt is added to the medium. In addition, the extreme functional variability of endothelial cells derived from different vascular regions of the brain may further confuse the final results. For instance, when the cells have an arterial, venous or capillary origin or when they derive from vessels of different brain regions.

The shared view of the reviewers is that, if the authors agree to drop these data (subsection “Rapid loss of the CNS EC transcriptional signature upon culturing of primary brain ECs” and Subsection “Changes in accessible chromatin in primary brain EC culture reflect loss of Wnt signaling” and related results), the paper would be more focused and solid and, overall, will add a lot to our understanding about the diversification of endothelial cells to distinct organ-specific identities.

We hope that the authors accept to drop this part of the results so that the paper will be acceptable for publication in *eLife*. This conclusion was reached after lively and motivated discussion among the reviewers and the Editors.

---

## [Author Response]

Thanks for submitting your manuscript to eLife. The two reviewers and I found the paper of interest and certainly important for the advancement of the field.However, as specified in detail in their comments the reviewers feel that the manuscript requires revision. In particular, the data on in vitro brain endothelial cell culture should be done in a more accurate way. For instance, the role of canonical Wnt in maintenance of the BBB characteristics in vitro should be analyzed in more detail as recommended. Both reviewers 2 and 3 ask for other relevant additions for a more adequate comparison of cultured and freshly isolated endothelial cells. Some relevant references are missing:Vanlandewicjk et al., 2018; Paolinelli et al., 2013 for in vitro BBB cell culture.Importantly, Vanlandewicjk et al., should be cited and discussed.The full reviews are also included for your reference, as they contain detailed and useful suggestions.Reviewer #2:This manuscript entitled "Transcriptional and epigenomic landscapes of CNS and non-CNS vascular endothelial cells" analyzes transcriptional and epigenomic differences between four distinct types of ECs (brain, lung, liver and kidney) using RNA-seq, DNA methylation profiles and accessible chromatin regions with ATAC sequencing. In addition, the authors isolate brain ECs from P7 brains and culture them for a week to identify potential changes in RNA-seq, DNA methylome and accessible chromatin between brain ECs in vivo and in vitro that may account for loss of BBB properties in vitro.Although, multiple studies have now been performed on identification of transcriptional differences between ECs in different organs (brain, lung, liver) and the concept that the vasculature is highly specialized for each organ is well accepted in the field of vascular biology, the study provides some novel and interesting insights into the biology of transcriptional specializations of ECs for distinct organs. This is a tour-de-force study that provides a detailed analysis of differences in DNA methylome and accessible chromatin between distinct cell types.The study suffers from some drawbacks that need to be addressed in greater detail to enhance the significance of the study.1) One of the major analyses is the comparison between brain ECs in vivo and in vitro. The authors show nicely that several genes that are activated by canonical Wnt signaling become downregulated within a week in culture. What specific culture conditions were used to culture mouse endothelial cells? Several key studies in the field of vascular biology have shown that addition of serum or ECGS in the media can affect the transcriptional landscape of ECs in culture. Therefore, the findings from the authors about the presence of differences are not surprising. However, can the authors perform a similar analysis using more defined conditions and various EC media to determine which conditions may provide the minimum divergence from the brain ECs? This will be highly valuable for vascular biologists and represent a major advancement in the field.

The CNS endothelial cells were cultured in Lonza’s EGM-2 MV medium [5% fetal bovine serum, VEGF, EGF, FGF, and IGF1]. This is now described in greater detail in the Materials and methods section. Importantly, there was no genetic or pharmacologic treatment to activate canonical Wnt signaling, as described in Paolinelli et al., (2013)in vitro. The suggestion of additional studies with cultured CNS endothelial cells would represent a logical extension of the experiments and analyses described here but doing those additional experiments would represent a major endeavor and would – in our view – be beyond the scope of the present study.

2) Does addition of Wnts or Wnt agonists to cultured brain ECs restores their properties to a certain extent or is the divergence in culture so extensive that is irreversible? When does this process become irreversible?

Yes, constitutively activating canonical Wnt signaling in cultured brain ECs gives them molecular and tight junction characteristics more in line with their in vivo counterparts (Paolinelli et al., 2013). As Paolinelli et al., showed, this can be done with Wnt ligands or Wnt-containing conditioned medium, but it was most effectively done by lentiviral transduction with a LEF/TCF derivative carrying the β-catenin transcriptional activation domain. We are not aware of any data that examines reversibility or irreversibility in cell culture, but earlier work from several groups (including ours) using Cre-lox mediated gene deletion or activation showed that the BBB remains plastic throughout life and that the BBB can be lost or restored in vivo in postnatal mice (e.g., Liebner et al., (2008); Wang et al., (2012)). To flesh out these issues, we have expanded our description in the Results section of what has been done previously.

3) Several studies have shown that brain or lung ECs in culture retain some tissue-specific identity and expression of certain genes in culture. What happens to some of the BBB-specific genes in cultured brain ECs? The authors do not show in detail BBB specific genes.

For the analysis of in vivo vs. cultured ECs we have added the following. The new version of Figure 10 shows a plot of gene expression changes for 20 BBB-specific genes. Additionally, the new Figure 10—figure supplement 2 shows the in vivo vs. in vitro expression data for 9 Wnt signaling related or regulatory genes (e.g. Apcdd1 and Axin2) and 12 genes that code for membrane and intracellular mediators of Wnt signaling. We have also added a new file (Supplementary file 9) that lists the BBB genes that are suppressed in cultured ECs relative to in vivo ECs. Finally, we have added new data on adult brain ECs in vivo to expand the comparison with cultured brain ECs. In the original submission, all of the in vivo ECs were obtained at P7. The new data show that there is little difference when the comparison is performed with P7 or with adult ECs.

4) Do lung ECs diverge to a "generic" EC when cultured for several days? It will be important to test this hypothesis and show the data for another cell type beside the CNS.

This is an interesting question for ECs from every organ, since distinctive organ-specific EC pattern of gene expression are presumably programmed (or at least influenced) by the tissue micro-environment. We have added several references to the existing data, although there is a dearth of published data addressing this question at the genome-wide level. However, we think that doing additional experiments to explore in vivo vs cultured EC properties beyond the basic characterization that we have performed with CNS ECs in culture would be a major undertaking and is beyond the scope of the current study.

5) In Figure 7, the authors show some specific new markers for tip cells or capillary-A, capillary-V markers. A recent study by Betsholtz has shown that there is a zonal expression of several endothelial genes. Therefore, these findings are not surprising. Have the authors validated any of these markers in particular tip cell markers? They only show DCC. It will be important to validate other genes.

Additional validation is an excellent suggestion. We have performed in situ hybridization analyses on developing mouse retinas at P5-P7, when the annulus of tip cells is spatially distinct from the rest of the vascular plexus. This analysis has validated 3 novel tip cell genes that were predicted from our scRNA-seq and this new data is presented in a new figure (Figure 9).

Reviewer #3:Sabbagh et al. compare the transcriptional and epigenetic landscapes of endothelial cells in different organs: brain, liver, lung, and kidney. This is the first study to perform both RNA-seq, MethylC-seq and ATAC-seq of ECs in different organs, and the study also includes some brain EC vascular zonation analyses (a novel and timely topic). Lastly, the study compares expression of freshly isolated ECs and in vitro ECs cultured 8 days, but flaws exist with the design of this final experiment (see below comments). While the present study is interesting and has some novelty overall, the study could be improved by addressing the comments below:1) In vivo and in vitro transcriptional and epigenetic data (Figure 9 and Figure 10) are generated from two different strains of mice at two different developmental ages (in vivo: Tie2-GFP mice at P7; in vitro: C57BL/6 mice at P28-56); then the authors conclude from this data that in vitro culture affects gene expression. For this comparison to be valid, the authors should similarly analyze in vivo and in vitro cultured cells from the same genetic mouse strain at the same age, and then determine the degree to which in vitro culture shifts gene expression. The time in culture is also important as it might affect the results. For instance, after how many population doubling times EC were studied? With increased timing in culture ECs might turn towards senescent-like phenotype. It would be really cool to show for how long they can sustain close to their in vivo determine profile. This is also why comparing mice at similar age may be important especially as a recent study in Nature 2018 suggests that some endothelial cells do not fit the A-V zonation pattern and exhibit high expression of ribosomal protein transcripts, which indicates that protein synthesis occurs throughout the A-V axis.

As noted above in our reply to reviewer #2, we agree that a more in-depth analysis of the in vivo vs. in vitro properties of CNS ECs would be very interesting and would represent a logical extension of the experiments and analyses described here but doing those additional experiments would represent a major endeavor in itself and would – in our view – be beyond the scope of the present study. That said, we agree that a comparison between adult CNS ECs in vivo and adult CNS ECs in vitro would be valuable. In the original submission, the comparison was with P7 ECs in vivo vs. adult ECs in vitro. Therefore, we have generated RNA-seq and ATAC-seq data sets from adult brain and repeated the in vitro vs. in vivo analyses with these datasets. The results are in close agreement with the P7 comparison and are shown in the new Figure 10 and Figure 11. See, for example, the heatmap in Figure 10B, the PCA in Figure 10C, and the expression profiles in Figure 10D.

2) The entire study uses 2 biological replicates consisting of 2 total number of mice for liver, kidney and lung and 4 total mice (2 mice pooled per replicate) for brain transcriptional and epigenetic data (except brain ATAC-seq that was conducted with 3 biological replicates). We think that 2 biological replicates may not be sufficient.

The question of how many replicates to use for NextGen sequencing studies has been discussed by various authors (e.g. Liu, Zhou and White, 2014). We find that RNA-seq, ATAC-seq, and MethylC-seq give very good reproducibility between pairs of biological replicates, as shown by the scatter plots and R-values which are shown for each pair of replicate data reported here. For example, the inter-replicate R-values for the 4 EC types ranges from 0.91 to 0.95, whereas the R-values between different EC types ranges from 0.45 to 0.77. We do not think that adding a third biological replicate will change the analysis.

3) The authors may consider developing a website to enable readers to more easily access the study's transcriptional and epigenetic data, as many other groups have done (see for example Zhang et al., 2014; He et al., 2016; Vanlandewijck et al., 2018).

This is a great suggestion, and we have now developed a user-friendly web site (using the ShinyApp platform) so that the scientific community can query our data. The link is: https://markfsabbagh.shinyapps.io/vectrdb/

4) The manuscript's overall interpretation, readability and data presentation could be improved. For example, the authors present the degree of methylation of differentially expressed EC genes in each organ expressed as gene browser snapshot and bar/scatter plots (Figure 2—figure supplement 2), but no statistical analyses are performed, and the methylation pattern interpretation is not always clear. Moreover, the many gene browser snapshots presented throughout the figures are not reader friendly and do not clearly demonstrate the point the authors are trying to make in the text.

Thank you for that comment. We have gone through the text to try to improve readability. Part of the challenge is simply that there are a lot of data and a lot of analyses. We have also modified some of the browser images by adding arrows (and referring to the arrows in the text) – this will make it easier for the reader to focus on the points of interest. Regarding the gene expression analyses in Figure 2—figure supplement 2C, each of these genes had been flagged in an initial comparison as differentially expressed at a statistically significant level. Furthermore, the DMVs indicated by colored bars in the browser images were called using appropriate statistical packages.

Regarding the barplot in Figure 2—figure supplement 2B, the reviewer is correct that there is no statistical analysis for the comparison between tissues because we could not figure out what a biological meaningful statistical analysis would consist of. The goal of this analysis was to highlight a trend that statistically significant DNA methylation valleys at statistically significant differentially expressed transcription factor genes exhibit differences in the size of the valley. Moreover, we think that the point is clear from inspection of the primary data: some TF genes (such as the ones shown in the top two rows of Figure 2D) overlap large regions of demethylation in all four EC types, and others (such as *Gata1, Spi1, Pou5f1* and *Tcf4* in the third row of Figure 2D) have essentially no demethylation. At present, all that we can say is that this is a striking observation in search of an explanation.

5) In addition to reporting Gene Ontology (GO) pathways shared by brain, liver, lung and kidney (Figure 2—figure supplement 1E), the authors should also include GO pathways terms that are unique to ECs in each organ.

We have added this analysis (Gene Ontology for the four sets of ECTSGs) as File 3.

6) The authors present heatmaps depicting gene expression and mean fraction of methylated CG in the gene body for genes within four HOX clusters presented anterior to posterior (Figure 3C). Many of the genes shown lack an association between high gene expression and high methylation or low gene expression and low methylation. What is the interpretation of this figure? This should be added and adequately discussed in the text.

We have expanded the Results section related to the Hox gene clusters to clarify the analysis, and we have added arrows to the genome browser images to more clearly connect text and figures. The reviewer is correct that the correlation between methylation and expression shows substantial variability across the four clusters – the correlation is imperfect. Rather, like other developmentally important TFs, the trend appears to be that expression of a HOX gene corresponds to increased methylation upstream of the gene body. For example, *Hoxa5* whichis expressed in liver, lung, and kidney ECs but not brain ECs exhibits a hypomethylated gene body in all ECs but a hypermethylated promoter region in only peripheral ECs. The heat maps in Figure 3 present the basic correlation between organ position along the rostro-caudal axis, HOX gene expression, and HOX gene body methylation, but our simple analysis does not reflect the regulatory complexities of the HOX clusters, which are still not fully understood. The lack of an apparent association between high gene expression and high methylation is likely due to the calculation of methylation in the gene body. We have added to Figure 3 a summary heatmap of total methylation for each of the four HOX clusters for each of the tissue-specific ECs.

7) It is quite surprising the authors do not cite a recent Nature paper that is the first to present transcriptional analysis of vascular zonation profiles of brain endothelial cells and mural cells (Vanlandewicjk et al., 2018). This paper should be cited and discussed. In regard to this paper and clustering of different brain endothelial types along the A-V axis – arterial, capillary and venous (also shown in the authors Figure 7 and Figure 8), do the authors know which endothelial cell clusters are represented by their in vitro analysis? This also could be the source for variation between in vitro and in vivo findings, as in vitro ECs might have differentially enriched clusters of cells compared to in vivo, depending on the source of origin.

The Vanlandewicjk et al., paper appeared the week before we submitted our manuscript, when our text was in essentially its final form. We decided at that time to submit our manuscript without expanding the Discussion section to include an assessment of the Vanlandewicjk et al., data. We are pleased to now have an opportunity to include that assessment in an expanded Discussion section.

Regarding the different properties of CNS ECs in vivo vs in vitro, we have explored the origins of the in vitro CNS ECs (capillary, artery, vein, etc.) that we analyzed by assessing the abundances of the top 500 transcripts from each of the EC clusters defined by scRNA-seq. This new analysis (Figure 10—figure supplement 1) shows that the in vitro ECs have a gene expression profile that is a good match to all six scRNA-seq clusters. For at least the first two days the cultured ECs retained a core attribute of the BBB: an ability to extrude puromycin, thereby rendering them selectively resistant and allowing us to kill all of the non-ECs (neurons, glia, pericytes, etc.) in the culture. The ECs were cultured for 8 days before harvesting.

8) The EC Methyl-seq correlation between different organs is very good: r=0.95-0.97 (Figure 2A; Figure 2—figure supplement 1B) and r=0.90-0.92 (Figure 4—figure supplement 3), compared to the EC RNA-seq (normalized counts) correlation between different organs (r=0.45-0.61) (Figure 1B). Why is this? The authors may want to elaborate on this in the discussion.

The higher correlation among methylomes largely reflects the inclusion of all genes (>20,000) in the comparison, most of which are not differentially expressed and not differentially methylated. By contrast, the RNA-seq comparison was limited to EC-enriched genes – thus, it did not include unexpressed genes or housekeeping genes. This is now described in the Results section.

9) It would be helpful if the authors can provide quantification for their immunohistochemistry data (e.g., Figure 6).

That is a good idea. We have done this, and it has been added to the Results section and shown in a new supplemental Figure (Figure 6—figure supplement 2).

10) This study only uses male mice. This is a limitation given the NIH's urge to be inclusive of both genders.

A good point. We used male mice to try to minimize any sex-dependent variability (which is likely modest compared to the inter-organ variability represented by BBB gene expression) and we could have just as easily used only female mice. This is a real conundrum because to do the studies in the most careful manner (as per the NIH) means analyzing both males and females and either pooling the data (with the attendant risks of increased biological variability) or doubling the number of mice, the amount of sequencing, and much of the costs. This highlights the advantages of working with a hermaphrodite like *C. elegans*!

[Editors' note: further revisions were requested prior to acceptance, as described below.]

Thank you for resubmitting your work entitled "Transcriptional and epigenomic landscapes of CNS and non-CNS vascular endothelial cells" for further consideration at eLife. Your revised article has been favorably evaluated by Didier Stainier (Senior Editor), a Reviewing Editor, and two reviewers.All of the reviewers agree that the paper has been improved significantly after revision. However, some doubts still remain that need to be addressed before acceptance. These are related to the comparison of brain derived endothelial cells in vivo or in culture conditions in vitro.Specifically, a more accurate study on the culture conditions used seems important to draw conclusions. It has been known for many years that when brain endothelial cells are isolated and cultured they lose their BBB properties quite rapidly, unless they are maintained in co-culture with astrocytes (see, for instance, papers from R Cecchelli et al., and many others) or when Wnt is added to the medium. In addition, the extreme functional variability of endothelial cells derived from different vascular regions of the brain may further confuse the final results. For instance, when the cells have an arterial, venous or capillary origin or when they derive from vessels of different brain regions.The shared view of the reviewers is that, if the authors agree to drop these data (subsection “Rapid loss of the CNS EC transcriptional signature upon culturing of primary brain ECs” and Subsection “Changes in accessible chromatin in primary brain EC culture reflect loss of Wnt signaling” and related results), the paper would be more focused and solid and, overall, will add a lot to our understanding about the diversification of endothelial cells to distinct organ-specific identities.We hope that the authors accept to drop this part of the results so that the paper will be acceptable for publication in eLife. This conclusion was reached after lively and motivated discussion among the reviewers and the Editors.

We are happy to comply with your and the reviewers’ judgment regarding manuscript changes, and we have, accordingly, removed the section of the manuscript (text and figures) dealing with cultured endothelial cells. The only substantive addition we have made is to note in the Results section that many ETS:ZIC paired motifs are found in the LTRs of a endogenous retrovirus family.